# Deep-seated gravitational slope deformation scaling on Mars and Earth: same fate for different initial conditions and structural evolutions

Olga Kromuszczyńska[1], Daniel Mège[2,3], Krzysztof Dębniak[1], Joanna Gurgurewicz[2], Magdalena Makowska[1] and Antoine Lucas[4]

[1]Institute of Geological Sciences, Polish Academy of Sciences, Research Centre in Wrocław, Podwale 75, 50-449 Wrocław, Poland
[2]Space Research Centre, Polish Academy of Sciences, Bartycka 18A, 00-716 Warsaw, Poland
[3]Laboratoire de Planétologie et Géodynamique, Université de Nantes, UMR CNRS 6112, 2 rue de la houssinière, Nantes, France
[4]Institut de Physique du Globe de Paris, Sorbonne Paris Cité, Université Paris, Diderot, Paris 75013, France

*Correspondence* to: Daniel Mège (dmege@cbk.waw.pl)

**Abstract.** Some of the most spectacular instances of deep-seated gravitational slope deformation (DSGSD) are found on Mars in the Valles Marineris region. They provide an excellent opportunity to study DSGSD phenomenology using a scaling approach. The topography of selected DSGSD scarps in Valles Marineris and in the Tatra Mountains is investigated after their likely similar postglacial origin is established. The deformed Martian ridges are larger than the deformed terrestrial ridges by one to two orders of magnitude, with however a similar height-to-width ratio, ~0.24. The measured horizontal spreading perpendicular to the ridges is proportionally 1.8 to 2.6 times larger for the Valles Marineris ridges than the Tatra Mountains, and vertically, 2.9 to 5.1 times, suggesting that starting from two different initial conditions, with steeper slopes in Valles Marineris, the final ridge geometry is now similar. Because DSGSD is expected to be now inactive in both regions, their comparison suggests that whatever the initial ridge morphology, DSGSD proceeds until a mature profile is attained. Fault displacements are therefore much larger on Mars. The large offsets imply reactivation of the DSGSD fault scarps in Valles Marineris, whereas single seismic events would be enough to generate DSGSD fault scarps in the Tatra Mountains. The required longer activity of the Martian faults may be correlated with a long succession of climate cycles generated by the unstable Martian obliquity.

## 1 Introduction

### 1.1 Deep-seated gravitational slope deformation

Deep-seated gravitational slope deformation (DSGSD) is a slow process that deforms mountain ridges, usually as a paraglacial process (e.g., Mège and Bourgeois, 2011) that is readily identified by uphill-facing normal fault scarps in the upper part of

ridge slopes, resulting in ridge-top splitting and summital valley development. DSGSD occurred in Valles Marineris, Mars, where it has been featured, though not analysed, on the earliest Viking images (Blasius et al., 1977), and identified as such much later (Peulvast et al., 2001). It is apparent at a glance that the scarp size is much larger on Mars than on Earth, but also that this difference scales with ridge size: much larger ridges have much higher scarps. This conforms to intuition, but there is

no obvious explanation for this. Intuition suggests a simple scaling law according to which larger ridges have a similar number of scarps than smaller ridges, but showing larger offsets, but why would extensional strain at larger ridges not be distributed over a larger number of scarps than for smaller ridges, with similar offsets? This question has been the trigger for this work, which quantifies this intriguing difference in DSGSD scarp height on Earth and Mars, and explores some implications.

In DSGSD, extension at the upper part of the ridge, in the absence of catastrophic slope failure, is associated with other deformation of the lower part of the ridge. In many instances it takes form of compressive bulging (e.g., Radbruch-Hall et al., 1976; Beget, 1985; Savage and Varnes, 1987; Chigira, 1992; Reitner et al., 1993; Ambrosi and Crosta, 2006; Hippolyte et al., 2006; Discenza et al., 2011). In some cases, the slope is observed to be overthrusting the valley (Mahr, 1977; Guerricchio and Melidoro, 1979; Bachman et al., 2009; Savage and Varnes, 1987), which suggests that a décollement may connect the faults

in the upper part of the slope with displacement at the bottom of the slope. In some other instances, the bottom of the slope is not deformed and the extension in the upper part of the ridge is absorbed within the ridge, the density of which is locally increased (Discenza et al., 2011), possibly by crushing (Beck, 1968; Mahr, 1977). Makowska et al. (2016) found that in homogeneous rock indeed, DSGSD in the upper part of the slope and at the bottom are disconnected, and that an internal core of intensely deformed rock underlies the uphill-facing scarps developed in the upper part of the slope. In this work we cannot

explicitly take lower ridge deformation into account, if any, due to systematic blanketing of basal ridge bedrock structure by a huge debris slope; nevertheless, the corresponding horizontal deformation that would result is accounted for implicitly by considering measurement of total ridge width where DSGSD is observed and where it is not.

### 1.2 The dissected Valles Marineris hillslopes

The Valles Marineris bedrock hillslopes show a dissected, "spur-and-gully" morphology at a glance similar to the morphology

of alpine mountains, locally degraded at the first order into tributary canyons and huge landslides (Lucchitta, 1978; Patton, 1991; Lucchitta et al., 1992). Spur-and-gully dissection has taken the form of subparallel, digitate spurs separated by gullies covered by long slope deposits (Lucchitta, 1978; Howard, 1989). The upper parts of the slopes are steeper than the lower parts (Lucchitta, 1978). The latter are covered by talus of material accumulated at the angle-of-repose (Patton, 1981). The spurs are usually perpendicular to the slope; however, in some areas, their oblique orientation relative to chasma trend suggests structural

control by vertical fractures oblique to the troughs that must be tens of kilometres long (Sharp 1973; Blasius et al., 1977). High resolution image data available since the early 2000's have revealed that outside these tributary canyons and huge landslides, the dissected, pristine morphology shows a variety of smaller-scale geomorphological features which are familiar to mountain geomorphologists, with the association or superimposition of a rich variety of landforms controlled by mountain

permafrost degradation, fluvial erosion, talus development, and gravity (Mège and Bourgeois, 2011; Gourronc et al., 2014; Dębniak et al., 2017). The difference with alpine mountain geomorphology on Earth is therefore not that much a matter of morphological details; it rather lies in the size of the landsystems. In the main troughs, the height of the Valles Marineris slopes is several kilometres, and up to 9 km, whereas typical mountain slopes on Earth are hundreds of metres to a few kilometres

high.

## 1.3 Deep-seated gravitational slope deformation in Valles Marineris

This work focuses on a series of morphotectonic features observed along Valles Marineris walls displaying the spur-and-gully morphology, including uphill-facing normal faults scarps and crestal grabens. Such features are systematically observed when spur-and-gully dissection occurs on internal ridges within the main Valles Marineris chasmata. They denote extensional

tectonics, but boundary forces that result in crustal "rifting" are unlikely to be the cause of this deformation even though rifting is frequently considered to have been a major contributor to the formation of some of the main Valles Marineris chasmata (e.g., Masson, 1977; Frey, 1979; Schultz, 1991, 1995a, 1998; Mège and Masson, 1996a, 1996b; Peulvast et al., 2001; Mège et al., 2017). One of the main reasons is that uphill-facing scarps and crestal grabens have not been reported to have formed in terrestrial rift zones on Earth nor are they expected to form in experimental models (e.g., Corti, 2012 for the East African Rift

System). In extreme cases, when crustal stretching has been on the order of hundreds of percent, cracking and normal faulting is pervasive in horsts as much as in grabens (Angelier and Colletta; 1983), which has certainly not been the case in Valles Marineris (e.g. Schultz 1991, 1995a; Mège and Masson 1996b; Mège et al. 2003; Andrews-Hanna, 2012).

Another reason is that when the basal ridge slope topography can be accurately studied (i.e. not mantled by debris aprons), normal faulting on the ridge crest and slopes appears to be counterbalanced by basal slope bulging (Mège and Bourgeois,

2011). Ridge-top splitting has been interpreted by Lucchitta et al. (1992) and Treiman (2008) as crustal zones strengthened by dyke intrusion or cemented by central fractures. Arguing that such a fracture origin is not well documented on Earth either, Mège and Bourgeois (2011), Kromuszczyńska et al. (2012) and Gourronc et al. (2014) interpreted these features as an effect of DSGSD. Uphill-facing normal faulting and crestal extensional deformation is indeed well documented on Earth in areas of DSGSD. In most described terrestrial instances, such as in the Alps of Europe, Japan, New Zealand, and the Andes, DSGSD

has been described in mountain ridges glaciated during the Quaternary (see a review in the supplementary material of Mège and Bourgeois, 2011), Such a postglacial context, in the Valles Marineris case, is adapted to the recently identified and widespread glacial landsystem (Gourronc et al., 2014; Mège et al., 2017; Dębniak et al., 2017), is also in agreement with expected slope deformation in this context (Makowska et al., 2016), and additionally provides a good framework (Mège and Bourgeois, 2011; Cull et al., 2014) to understand the detected mineralogical occurrences as from CRISM (Roach et al., 2010;

Cull et al., 2014). Postglacial deformation, as far as DSGSD is concerned, includes the paraglacial response of rock slopes to changing stress conditions after glacier retreat (e.g., Ballantyne, 2002), i.e. hillslope debuttressing, but also some longer term deformation made possible or facilitated by slope debuttressing. This includes upslope migration of stress release, and

postglacial water flow, pressure variations and subcritical failure occurring for instance within the framework of the geologic response to climate cyclicity (Pánek et al., 2017).

In this work, we will therefore assume that DSGSD is the most likely origin for the uphill-facing scarps and crestal grabens observed on the slopes of the Valles Marineris spur-and-gully ridges. We also consider that a postglacial origin is likely (Mège and Bourgeois, 2011; Gourronc et al., 2014), given its consistency and adaptation to the evidence of a widespread Valles Marineris glacial landsystem subject to short-period climatic variations (Laskar et al., 2004; Levrard et al., 2004) under which additional slope morphologies may develop (e.g., Quantin et al. 2005; Chojnacki et al., 2016).

## 1.4 Scaling of processes involved in deep-seated gravitational slope deformation

Some gravity processes do not similarly develop at different scales. For instance, landslide propagation does not critically depend on the same parameters when small or large. Landslides that are small with respect to mountain size are influenced by the surrounding mountainous topography, the first effect of which being the development of an inclined transport channel that overlies an accumulation of debris on a less inclined slope. An example is the Thurwieser landslide in the central Alps (e.g., Sosio et al., 2008). For landslides that involve a large fraction of the mountain slope, the deposits spread in the nearly flat valley or plain downstream, such as in the case of the Socompa rock avalanche in Chile (e.g., Kelfoun and Druitt, 2005), the Blackhawk landslide in California, United States  (e.g., Shreve, 1987), and the Frank rock slide in Alberta, Canada (e.g., Daly et al., 1912). Although similar at first order, landslide propagation is not controlled by the same parameters because the landslide slope is different. It was shown that landslide volume is one of the factors that control landslide propagation (McEwen, 1989; Lucas et al., 2014; Johnson and Campbell, 2017) when the slope angle of the propagation plane is steeper than ca. 20º (Farin et al.; 2014; Borykov et al., submitted), whereas it has no influence for more gentle slopes.

This dependency of landslide propagation on slope, and indirectly on volume (and friction), initially identified on laboratory experiments (Farin et al., 2014), could be adequately documented by natural examples thanks to some very large Martian landslides (Johnson and Campbell, 2017; Borykov et al., submitted), much larger than any terrestrial landslide, which help populate the landslide dataset for voluminous landslides that propagate on nearly flat surfaces. Similar to landslides, DSGSD occurs on mountain slopes that are much smaller on Earth than on Mars, and at the first order, their origin may be similar. However, like in the case of landslides, the difference in scale may be associated with differences in controlling parameters. We will not answer this complex question in this work, but instead provide quantitative information that may help in future works aiming at constraining the parameters that control DSGSD over a broad range of sizes.

## 1.5 Scaling with the Tatra Mountains

We compare DSGSD in Valles Marineris with DSGSD in the Tatra Mountains in Slovakia and Poland (Fig. 1), a historic region of DSGSD investigations (Jahn, 1964; Nemčok, 1972; Mahr, 1977). Investigating this potential analogue area is motivated both by some similarity in DSGSD development conditions, and easy field access.

DSGSD develops independent of the structure and lithology of the topographic ridge, although major strength contrasts affect the strain field and therefore the expected observed deformation (e.g., Makowska et al., 2016). Such contrasts may be provided by weak/strong rock contrasts and major faults. In Valles Marineris, the upper part of the slopes are thought to be volcanic, akin to trap series (McEwen et al., 1999. Beyer and McEwen, 2005), with thin lava layering in which jointing is pervasive.

This level may be viewed as a typical homogeneous rock mass (Hoek, 1983; Schultz, 1995b). The ridge material that makes the lower part of the ridges is usually covered by debris slopes. Nevertheless, when they are observed they appear to be made of a massive basement rock or magmatic intrusives (Williams et al., 2003). In the Tatra Mountains, the measured scarps are located in a granodioritic intrusive body (Nemčok et al., 1994). In both Valles Marineris and the studied sites of the Tatra Mountains, the role of the tectonic fabric in the development of the DSGSD features is therefore expected to be minor, although

elsewhere in the Tatra Mountains a tectonic breccia level parallel to DSGSD displacement has been suspected to have influenced scarp development (Pánek et al., 2017).

In the Tatra Mountains, deglaciation was ended at ~8500 B.P. (Lindner et al., 2003), and the ages obtained for DSGSD in the Polish and Slovakian Tatras are between 15.7-4.3 ka (Pánek et al., 2017). Some DSGSD features in the Tatra Mountains started soon after local deglaciation, but the influence of climatic cycles and their geological consequences, especially in terms of

groundwater flow, pore pressure variations, and their implications for rock corrosion, is suspected to be major because in some instances scarp development occurred well after deglaciation. In Valles Marineris, the glacial landsystem cannot be dated with a similar accuracy, and based on observations by Mège and Bourgeois (2011) it is not possible to exclude several or many periods of glaciations since the upper Hesperian. Martian climate instability actually favours the interpretation of many glaciation and deglaciation cycles, a response to incessant planetary obliquity variations (Laskar et al., 2004). For instance,

climate oscillations were shown to proceed with a rate as fast as 120,000 year per cycle in the last 10 million years (Levrard et al., 2004). Development of DSGSD features in Valles Marineris is therefore likely to have proceeded under a long succession of contrasted environmental conditions, perhaps resulting in a variety of processes operating recurrently, both short-term processes such as ridge postglacial debuttressing, and long-term evolution under the influence of liquid water in a thawing mountain permafrost (e.g. Noetzli and Gruber, 2009; Huggel et al., 2013).

DSGSD in Valles Marineris and the Tatra Mountains are probably not active anymore, being dated Hesperian to Lower Amazonian for Valles Marineris (Mège and Bourgeois, 2011) and before the Late Quaternary for the Tatras (Panek et al., 2017), therefore while comparing both regions, we are comparing finite deformation in ridges assumed to be akin to homogeneous rock masses at the first order.

## 2 Data and methods

**2.1 Data**

The observations of the Valles Marineris trough system (Fig. 1a) reported here were made using Mars Reconnaissance Orbiter/CTX imagery as a baseline. Higher resolution images from Mars Reconnaissance Orbiter HiRISE were used for more

detailed research when available. For topography, the resolution of the HRSC digital elevation model (DEM) mosaic of Valles Marineris (Gwinner et al., 2009) is too coarse for this work, whereas HiRISE DEMs would be adapted but were either not available in the study sites, or did not cover enough surface to be used. DEMs were generated from CTX stereo pairs using SocetSet®, and have the appropriate vertical precision of ca. 15 m, and were used to extract topographic profiles. Nevertheless,

some DSGSD sites (including e.g. in Ius Chasma) could not be investigated due to the poor geometry of the CTX images available during the lifetime of this project, which did not make possible generation of good quality DEMs.

DSGSD scarps were studied in the Tatra Mountains during two periods of field work. Elevation data in the Tatra Mountains were collected using the GPS device Garmin GPSMap 62s in differential (WADGPS) and non-differential mode, as explained in Kromuszczyńska et al. (2016), during two field campaigns conducted in September 2012 and June 2013 in the Higher Tatra

Mountains in Poland and Slovakia, and the Lower Tatra Mountains in Slovakia.

**2.2 Methods**

The deformation produced by DSGSD on Mars and Earth are compared by quantitative interpretation of topographic profiles. Profile generation and analysis is done with ArcGIS 3D Analyst. The topographic profiles are then analysed in a graphic software.

On the basis of the DEMs (in case of Martian study sites) and field observations (in case of terrestrial study sites), as well as the profiles themselves, the mean local ridge slopes are identified and marked. Then, the faults cutting the profiles are located and marked as two lines illustrating two mean fault dip angles, $\alpha = 60º$ and $\alpha = 70º$. The lower value corresponds to theoretical normal fault angles classically found in shear experiments and theory (e.g., Cloos, 1932; Anderson, 1951); the higher corresponds to normal fault angles that are commonly found in extensional tectonic regimes a few tens of metres below the

surface (e.g., Gudmundsson, 1992), and could be similar to normal fault angles of DSGSD faults below the scarps. The 10º angle interval between 60° and 70° is considered as a plausible range of angular values, which cannot be retrieved from topography due to fault scarp erosion and debris accumulation. The trace of DSGSD scarps in map view both in Valles Marineris and in the Tatras sites indicate that vertical fractures play a negligible role in ridge deformation.

Fault mechanics equations do not scale with gravity, implying that similar angles are expected on Martian rocks than in

terrestrial rocks. The horizontal displacement $x$ and vertical displacement $z$ on DSGSD fault scarps are measured as on Fig.2. The measured fault displacements are scaled in a second step. This allows to compare the scarps between the 6 study sites. The scaled horizontal displacement is calculated by dividing the average value $\bar{x}$ of horizontal fault displacement $x$ measured on the ridge by the width $L$ of the ridge:

$$Dh = \bar{x}/L \tag{1}$$

The scaled vertical displacement $Dv$ is obtained by dividing the average value $\bar{z}$ of vertical fault displacement $z$ measured on the ridge by the height of the ridge $H$:

$$Dv = \bar{z}/H \tag{2}$$

The ratio $R$ of ridge height $H$ to width $L$, or aspect ratio, of the studied ridges allows to examine the similarity in the shape of the ridges as observed today, after DSGSD:

$$R = H/L \tag{3}$$

In order to compare the finite deformation of ridges, the maximum displacement found on each site is used, and is inferred from the maximum displacement measured along each profile:

$$\Delta x = \max\left(\sum_{i=1}^{n1} x_{p1}^i, \dots, \sum_{i=1}^{nN} x_{pN}^i\right) \tag{4}$$

and

$$\Delta z = \max\left(\sum_{i=1}^{n1} z_{p1}^i, \dots, \sum_{i=1}^{nN} z_{pN}^i\right) \tag{5}$$

where $\Delta x$ is horizontal strain and $\Delta z$ is vertical strain for each site, respectively, containing $n$ scarps along a given $p1, \dots, pN$ profile.

In terms of strain, the maximum strain found on each site is:

$$\varepsilon_x = \Delta x/L \tag{6}$$

and

$$\varepsilon_z = -\Delta z/H \tag{7}$$

**3 Study sites**

Three sites with clearly visible DSGSD features were selected in Valles Marineris, M1 to M3, based on DEM generation possibility, and three in the Tatra Mountains, T1 to T3 (Table 1).

**3.1 Valles Marineris**

Study Site M1 is located in the westernmost part of Coprates Montes, in Coprates Chasma (Fig. 3a). DSGSD there takes a form of crestal graben, and uphill-facing normal fault scarps on the northern side of the ridge. The western and eastern parts of the first study site are separated by a 26 km wide landslide alcove, with a scar aligned on both sides with the scarps of DSGSD crestal graben. Site M2 is the ridge separating Melas and Candor chasmata (Fig. 3b). Eastward, the ridge splits at the middle and lowers eastward almost down to the chasma floor, interpreted as a consequence of increased crestal extension by Mège and Bourgeois (2011), then reinterpreted as a glacial valley developed along a former crestal graben by Gourronc et al. (2014). Uphill-facing normal fault scarps developed on both ridge sides. Site M3 is a ridge located between Candor and Ophir chasmata (Fig. 3c). DSGSD features are represented by distinct crestal graben, which cause the ridge-top to split. On the slopes uphill-facing normal fault scarps are slightly visible. The global ridge parameters are given in Table 1.

## 3.2 Tatra Mountains

Sites T1 and T2 are located in the Slovakian Tatra Mountains, and Site T3 in the Polish Tatras. Site T2 (Fig. 4a) contains DSGSD features on a ridge east of Jamnícke Sedlo and Ostrý Roháč summit. A few uphill-facing normal faults scarps cut the ridge on its both slopes. The crest is wide and clearly reworked by DSGSD activity. The height of individual scarps reaches
up to 5-10 m. Site T2 (Fig. 4b) is a ridge west of Veľká Garajova Kopa on the way to Veľká Kopa. In the west, DSGSD features express as uphill-facing normal fault scarps distributed over the whole slope and associated with soil creep. In the east, the crest becomes narrower, uphill-facing normal fault scarps are restricted to close to the crest, and ridge-top splitting occurs by way of fresh, 20-cm wide tension fractures. In this area, a few uphill-facing normal faults scarps cut the ridge on both sides. The crest is wide and clearly reworked by DSGSD activity. The height of the scarps reaches 5-10 m. Site T3 (Fig.
4c) includes DSGSD features on the ridge south of the Ornak summit. The ridge has a wide crest cut by a series of normal faults on both sides that generated high uphill-facing scarps. The height of the highest scarps exceeds 10 m.

## 4 Results

### 4.1 Fault displacement observed along topographic profiles

On Mars, measurements were done along 11 topographic profiles in Site M1 (Fig. 6a). In sites M2 and M3, 5 and 6 profiles
were used, respectively (Fig. 6b-c). On Earth, 9 profiles were obtained in Site T1 and 13 in sites T2 and T3 (Fig. 7a-c). Profiling perpendicular to the ridges was conducted along crest lines, resulting in broken profile traces, and along straight lines. In a given site, these two methods yielded similar results. As examples, profiles following spur-and gully crest lines are provided for the sites M1 and M3. All the fault displacement measurements are provided as tables in Supplementary Material 1.

Two interpretations have been done of the central valley of Site M2, which have implications for DSGSD measurement. If it is a central graben similar to the central graben at, e.g., Site M1 that underwent further extension, as interpreted by Mège and Bourgeois (2011), the valley depth is a proxy to the crestal graben height. Alternatively, if it is a glacial valley (Gourronc et al., 2014), then there is no crestal graben observed. In that case, it is likely that a crestal graben of unknown depth existed, and guided the orientation of the glacial valley before being fully ablated. For this reason, profiles for Site M2 consider two options:
one with 6 normal fault scarps (consistent with Mège and Bourgeois, 2011), and one with the 3 southernmost and the northernmost scarps only (consistent with Gourronc et al., 2014). Both options represent end-members in which DSGSD in the site is maximised and minimised.

In the Tatra Mountains, the profiles were recorded along straight profiles, except in a few cases where vegetation made necessary deflection of the walking path, with a measured vertical precision of 40 cm (Kromuszczyńska et al., 2016).

Examples of interpreted profiles are given on Fig. 8. Supplementary Material 2 includes all the profiles. In sites M1 and M3, the largest displacements are found to have occurred along crestal graben faults. The mean measured displacements are reported in Table 2.

## 4.2 Fault displacement scaling with ridge dimensions

The scaled displacements are presented on Table 2 and graphically on Fig. 9. Although displacement along the fault scarps are two orders higher on Mars than on Earth, once scaled to ridge dimensions (eqs. 1 and 2), this difference becomes much less prominent (Fig. 9a-b). $Dh$ for the Valles Marineris sites (0.005–0.010) is 2 to 2.6 times the values for the Tatras sites (0.002–0.005) only. Due to the steepness of normal faults, this difference is larger vertically, with $Dv$ 2.6 to 5.1 larger for Valles Marineris (0.056–0.204) than for the Tatra Mountains (0.20–0.34).

In Site M2, $Dv$ is much higher than in the other Valles Marineris sites, whatever the interpretation of the morphology (Fig. 9b, Site M2a or M2b). This is interpreted as a consequence of glacial abrasion of the highest part of the ridge, as discussed in Section 5.2. $Dv$ is therefore overestimated by an unknown amount.

Why is DSGSD fault displacement on Mars of the same order as on Earth once scaled? The smaller gravitational acceleration at the surface of Mars (3.71 m·s$^{-2}$) than at the surface of Earth (9.81 m·s$^{-2}$) tends to increase the stability of Martian topographic

ridges compared to terrestrial ridges having similar height and slope angles (Makowska et al., 2016). Conversely, the much more voluminous ridges on Mars tend to build a higher gravitational potential, and make Martian ridges more deformable. These results show that both effects tend to compensate.

## Discussion

## 5.1 Fault strain distribution

The very large fault offsets measured on individual faults in Valles Marineris require cumulated events (e.g., Fossen, 2010, p. 172–174). The gradual dominance of slip along master faults with deformation time in a given fault set, at the expense of small faults, is a consequence of fault linkage with growing deformation (Cartwright et al., 1995). Because DSGSD is observed at the surface and does not extend deep below the base level of the deforming ridge (e.g., Makowska et al., 2016), the normal stress is small, which makes stable sliding more likely than stick-slip sliding (Marone and Scholz, 1988). Larger fault slip in

Valles Marineris is therefore made possible by plastic ridge deformation over a time span longer than the deformation time of the Tatra Mountains ridges. The valley glaciers expected to have occupied Valles Marineris chasmata floors are larger than the valley glaciers in the Tatra Mountains by at least two orders (Mège and Bourgeois, 2011; Gourronc et al., 2014), ridge slope deglaciation may have taken a long time, promoting long-lasting fault slip during DSGSD. Furthermore, the chaotic orbital obliquity regime of Mars (Laskar et al. 2004) makes realistic a very large number of glaciation and deglaciation cycles in

Valles Marineris throughout the history of Mars, and as many glacier advance and retreat cycles generating incremental DSGSD-induced fault displacements. In contrast, the paraglacial structures in the Tatra Mountains are not expected to be older

than 400 ka (Lindner et al., 2003). Clay gouge infilling (Treiman, 2008) of possible pre-existing fractures along the ridges inherited from the formation of Valles Marineris (Schultz, 1995a; Mège and Masson, 1996a, 1996b) may have also promoted stable sliding (Fossen, 2010) under the new glacial loading conditions.

## 5.2 DSGSD dependency on ridge scale

The ridge aspect ratio $R$ (eq. 3) is constant for all or most of the ridges (depending in which situation, 2a or 2b, is the Melas-Candor ridge, as discussed below). Considering that all the ridges are now inactive, it follows that in spite of the different scale, they attained a similar final stage, making $R$ an estimate of DSGSD maturity *de facto*, with $R=0.24$ (Figure 8c) for a mature ridge. The range of $R$ is narrow (0.18–0.29) for Earth and Mars if the glacial valley at Site M2 is fully erosional (Fig. 8c, Site M2b), as interpreted by Gourronc et al. (2014). $R$ values are much more scattered (0.08–0.29) and unusual compared with the two other Martian sites if the central valley in Site M2 is of DSGSD origin only (Fig. 8c, Site M2a), as interpreted by Mège and Bourgeois (2011). Here, for the following reason we favour Gourronc et al. (2014), which results in a nearly constant value of $R$ for all the studies ridges.

Geomorphologic analysis of the Candor-Melas ridge shows that the erosional processes that shaped the northern side of the ridge are distinct for the processes that generates the spur-and-gully morphology on the southern side. Mège and Bourgeois (2011) found this unusual but did not succeed in interpreting it further than an advanced stage of DSGSD. The northern side of the ridge was taken as an unusual example of extreme development of DSGSD. Three years after, the same group reinterpreted the chasmata around this feature as part of a huge glacial landsystem, based on many and widespread geomorphological observations (Gourronc et al., 2014). Within the framework of a regional glacial landsystem, the northern side of the Melas-Candor ridge would naturally form by glacial erosion proceeding eastwards from a narrow graben located west of the ridge crest. The graben would widen and deepen, eventually resulting in a structurally controlled U-shaped valley and abrasion and the ridge crest. Mège and Bourgeois (2001) had therefore hypothesized (2a) whereas Gourronc et al. (2014) corrected this view by proposing (2b). Interpretation (2a) is therefore still considered, although we now favour our more recent interpretation (2b).

An implication of favouring interpretation (2b) is that the range of the difference between scaled horizontal and scaled vertical displacements for the studied Martian and terrestrial instances, already estimated in Section 4.2 from Table 2, is refined to 1.8–2.6 for *Dh* and 2.9–5.1 for *Dv*.

An implication of the similar aspect ratio of all the profiles is that DSGSD has evolved from initial geometries that may have been different but eventually converged toward a common final and stable shape, in which the ridge aspect ratio is ~0.24. DSGSD therefore tends to stop to a stabilised state that is not scale dependent.

## 5.3 Initial ridge geometries

Analysis of the cumulated fault displacements (Table 3) can be used to retrieve information on the initial ridge geometries. The total horizontal extension (eq. 4) and vertical contraction (eq. 5) in the Tatras ridges is of the order of tens of meters,

corresponding to total horizontal elongation (eq. 6) and vertical shortening (eq. 7) of 0.8 – 1.2% horizontally and 8.6 – 16% vertically. In Valles Marineris, deformation is two orders larger, with total horizontal elongation 2 – 7% and vertical shortening of 21 – 52%. The proportionally larger deformation in Valles Marineris than in the Tatras, in spite of similar ridge aspect ratio, suggests that DSGSD evolved to its finite stage from an initially different aspect ratio. The collected measurements therefore

supports the proposition made in Section 5.2 that the *R* parameter may measure ridge maturity.

**Conclusion**

The ridges displaying DSGSD features in Valles Marineris are larger than the ridges displaying DSGSD in the Tatra Mountains by one order. In both regions, DSGSD is thought to have come to an end. Their aspect ratio (height/width) is similar, ~0.24 (Table 1). However, proportionally, horizontal displacement of the Valles Marineris ridges is three times higher than at the

Tatra Mountains ridges, implying that the initial slopes of the Valles Marineris ridges were higher than those of the Tatras ridges. It may be suggested that on the one hand, the maturity or immaturity (instability) of ridges affected by DSGSD may be inferred from their aspect ratio. On the other hand, this final, stabilised ridge geometry does not carry an indication on the initial shape of the ridge itself an information that may be retrieved by collecting information from individual DSGSD scarps. Individual fault displacements across DSGSD scarps in the Tatra Mountains are similar to fault displacement in most DSGSD

sites on Earth (see references in the Supplementary Table 1 of Mège and Bourgeois, 2011), suggesting that this conclusion may be extrapolated to other regions. Nevertheless, similar analyses need to be conducted in other ridges affected by DSGSD, formed in post-glacial as well as non-postglacial conditions, both inactive and active, before general conclusions can be drawn. DSGSD in the Tatra Mountains and Valles Marineris sites studied here occurred in rocks that at the first order may be considered as initially mechanically homogeneous. Rheological contrasts such as provided by lithologic contrasts or tectonic

fabric may however be critical parameters in the control of the geometry of a ridge subject to DSGSD (e.g., Makowska et al., 2016), and its evolution. The conclusions drawn here will probably be modified if the ridge structure departs too much from the homogeneous rock mass assumed here.

Fault displacement homothety implies that fault growth in both regions may not have been similar. Fault growth in the Tatras is consistent with single seismic events (yet to be formally identified in the field) for each scarp, whereas many events are

required to explain the large offsets measured across the Valles Marineris DSGSD fault scarps. Fault reactivation may have occurred as a geologic response to the long succession of glacial/interglacial cycles expected to have occurred throughout the history of Mars from celestial mechanics (Laskar et al., 2004) and in particular, in Valles Marineris (Mège and Bourgeois, 2011; Gourronc et al., 2014). In summary, although currently of globally similar geometry, the ridges affected by DSGSD in Valles Marineris and in the selected sites of the Tatra Mountains did not have the same initial conditions, nor is their structural

evolution similar.

**Tables**

| Site ID | Site name | Coordinates (central point) | $H$ (m) | $L$ (m) | $R$ |
|---|---|---|---|---|---|
| M1 | Coprates Montes | 67º54'W; 12º18'S | 5 500 | 22 000 | 0.25 |
| M2 (a) | Melas-Candor ridge | 73º55'W; 8º00'S | 5 000 | 56 000 | 0.08 |
| M2 (b) | Melas-Candor ridge | 73º55'W; 8º00'S | 5 000 | 30 000 | 0.25 |
| M3 | Candor-Ophir ridge | 73º25'W; 4º56'S | 5 700 | 35 000 | 0.18 |
| T1 | Jamnícke sedlo | 19º46'06''E; 49º12'02''N | 250 | 900 | 0.28 |
| T2 | Veľká Garajova kopa | 19º58'37''E; 49º12'05''N | 200 | 750 | 0.27 |
| T3 | Ornak | 19º50'11''E; 49º12'33''N | 420 | 2 200 | 0.19 |

Table 1: Ridge location and parameters in the three Valles Marineris sites and the three Tatra sites. The reason for distinguishing between 2(a) and 2(b) is given in the Section 4.

| Site ID | $\bar{x}$ (m) | | $\bar{z}$ (m) | | $Dh$ | | $Dv$ | |
|---|---|---|---|---|---|---|---|---|
| | $\alpha = 60°$ | $\alpha = 70°$ | $\alpha = 60°$ | $\alpha = 70°$ | $\alpha = 60°$ | $\alpha = 70°$ | $\alpha = 60°$ | $\alpha = 70°$ |
| M1 | 174 | 115 | 307 | 323 | 0.008 | 0.005 | 0.056 | 0.059 |
| M2(a) | 569 | 383 | 952 | 1022 | 0.010 | 0.007 | 0.190 | 0.204 |
| M2(b) | 421 | 289 | 726 | 794 | 0.008 | 0.006 | 0.158 | 0.173 |
| M3 | 312 | 214 | 538 | 584 | 0.009 | 0.006 | 0.094 | 0.102 |
| T1 | 4.5 | 3.1 | 7.9 | 8.5 | 0.005 | 0.003 | 0.032 | 0.034 |
| T2 | 2.3 | 1.6 | 3.9 | 4.3 | 0.003 | 0.002 | 0.020 | 0.022 |
| T3 | 6.5 | 4.4 | 10.3 | 11.1 | 0.003 | 0.002 | 0.025 | 0.026 |

Table 2. Mean horizontal and vertical displacements, in meters and scaled, at the six study sites. M2(a) assumes that the central valley in Site M2 is of purely DSGSD origin, whereas M2(b) assumes that the central valley has a purely glacial origin.

| Site ID | $\Delta x$ (m) | | $\Delta z$ (m) | | $\varepsilon x$ | | $\varepsilon z$ | |
|---|---|---|---|---|---|---|---|---|
| | $\alpha = 60°$ | $\alpha = 70°$ | $\alpha = 60°$ | $\alpha = 70°$ | $\alpha = 60°$ | $\alpha = 70°$ | $\alpha = 60°$ | $\alpha = 70°$ |
| M1 | 1250 | 840 | 1740 | 1790 | 0.06 | 0.04 | -0.32 | -0.33 |
| M2(a) | 3670 | 2480 | 3980 | 4240 | 0.07 | 0.04 | -0.80 | -0.85 |
| M2(b) | 1370 | 940 | 2350 | 2580 | 0.05 | 0.03 | -0.47 | -0.52 |
| M3 | 1250 | 840 | 1220 | 1350 | 0.04 | 0.02 | -0.21 | -0.24 |
| T1 | 16.4 | 10.9 | 28.2 | 29.9 | 0.02 | 0.01 | -0.11 | -0.12 |
| T2 | 15.6 | 11.5 | 27.2 | 31.4 | 0.02 | 0.02 | -0.14 | -0.16 |
| T3 | 26 | 18 | 36 | 38 | 0.012 | 0.008 | -0.086 | -0.090 |

Table 3: Ridge deformation in response to DSGSD in Valles Marineris and the Tatra Mountains. $\Delta x$: cumulated horizontal ridge elongation; $\Delta z$: cumulated vertical shortening; $ex$: horizontal strain (horizontal displacement normalized to ridge width in Table 1); $\varepsilon z$: vertical strain (vertical displacement normalized to ridge height), respectively. Elongation is positive and shortening is negative. 2(a) assumes that the central valley in Site 2 is of purely DSGSD origin, whereas 2(b) assumes that the central valley has a purely glacial origin. Hypothesis (b) is considered to be more likely (see Section 5.2).

**Figure captions**

Figure 1: Location of the study sites: (a) Valles Marineris, MOLA-based shaded relief map; (b) Tatra Mountains, SRTM-based shaded relief map.

Figure 2: Parameters used for calculation of vertical and horizontal displacement across (a) topographic ridge and (b) uphill-facing normal fault scarp. The black lines indicates pre-DSGSD ridge topography and the red lines, post-DSGSD topography. H is observed ridge height, L is observed ridge height, α is fault dip angle, x is horizontal fault displacement, and z is vertical fault displacement.

Fig. 3. Martian study sites. White dot indicates landmark for easier site localisation, with coordinates in the Mars Sphere datum. (a) Site M1: Coprates Montes. DSGSD is observed on the ridge on both sides of the landslide. Landmark is at 11°52'29.3"S, 68°19'44.1"W; (b) Site M2: Melas-Candor boundary ridge. Landmark is at 8°29'4.9"S, 72°36'25.1"W; (c) Site M3: Candor-Ophir boundary ridge. Landmark is at 4°46'33.6"S, 73°45'57.8"W. The boxes locates the areas where the profiles were measured (Fig. 6). Arrows indicate DSGSD scarps. MRO/CTX mosaic draped on Mars Express/HRSC topography.

Figure 4: Tatra study sites and location of field photographs (Figure 5). Lines are fault scarps, dashes indicate the downthrown block. For better readability, or where the scarp orientation is not well defined they are not always drawn. White dot indicates landmark for easier site localisation, with coordinates using the WGS84 datum. (a) Site T1, east from Jamnícke Sedlo. Landmark is at 49°12'04.5"N, 19°45'48.0"E; (b) Site T2: a ridge west of Veľká Garajova Kopa. Landmark is at 49°12'09.0"N, 19°58'59.0"E; (c) Site T3: ridge south from Ornak. Landmark is at 49°12'46.5"N, 19°50'08.5"E. Imagery is from ESRI World Imagery; contours are extracted from ASTER GDEM. ASTER GDEM is a product of METI and NASA.

Figure 5: Field photographs of the Tatra sites (located on Figure 4): (a) Site T1; (b) Site T2; (c) Site T3. Arrows indicate some of the DSGSD scarps.

Figure 6: Martian sites, location of the analysed fault scarps (red), CTX DEM (white boxes) and profiles (numbered white lines) used for fault displacement analysis. (a) Site M1; (b) Site M2; (c) Site M3. MRO/CTX mosaics realised with JMARS (Christensen et al., 2009).

Fig. 7. Terrestrial sites, location of the profiles. (a) Site T1, across a crest line east from Jamnícke Sedlo; (b) Site T2, the ridge west from Veľká Garajova Kopa; (c) Site T3, a ridge south from Ornak. GoogleTM Earth; 2015 CNES/Astrium.

Figure 8a: Determination of fault traces (continuous lines) from topographic slope around fault (dotted line) measured at the three Martian study sites, assuming fault dips of 60° (red) and 70° (blue).

Fig. 8b. Determination (dotted lines) of fault traces (continuous lines) measured at the three Tatra study sites, assuming fault dips of 60° (red) and 70° (blue).

Figure 9: Scaled horizontal displacement $Dh$ (a), scaled vertical displacement $Dv$ (b), and height to width ratios R (c) in the study sites. For Site M2, two end-members fault displacement interpretations of the Melas Chasma - Candor Chasma central depression are given: option (a) corresponds to the fully tectonic interpretation proposed by Mège and Bourgeois (2011) and option (b) to a mixed tectonic/glacial interpretation (Gourronc et al., 2014). Dashed line in 9c indicates the average height-to-width ratio when the option (b) of site M2 is considered.

**Supplementary material**

Supplementary Material 1: Fault scarp measurement data for the six study sites.

Supplementary Material 2: Extensive set of topographic profiles for the six study sites.

**Acknowledgements**

This work has been financially supported by the Foundation for Polish Science, project FNP TEAM/2011-7/9 "Mars: another planet to approach geoscience issues" and the OPUS/V-MACS project no. 2015/17/B/ST10/03426 of the National Science Centre, Poland. ASTER GDEM is a product of METI and NASA. Both reviewers are thanked for help in methodological clarification and in improvement of many details.

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

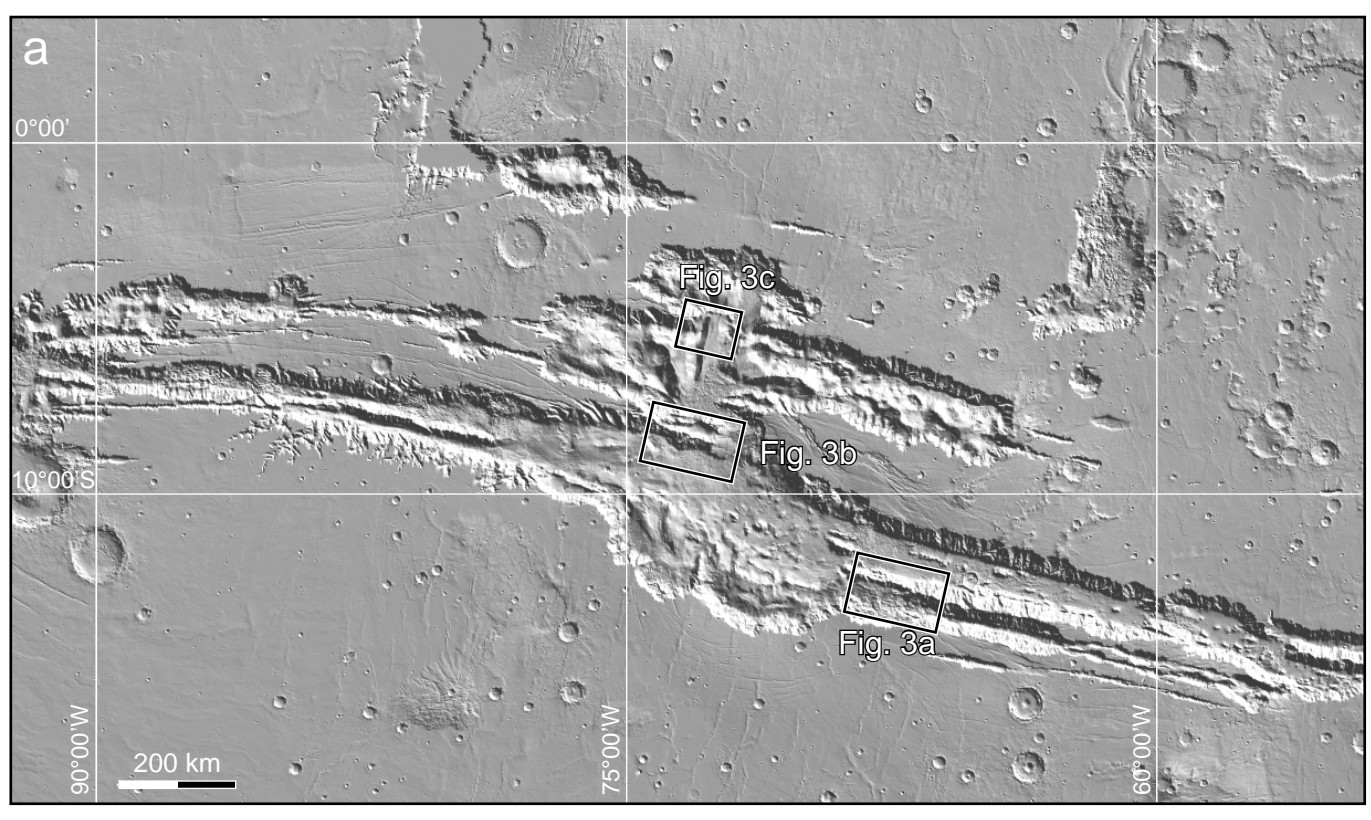

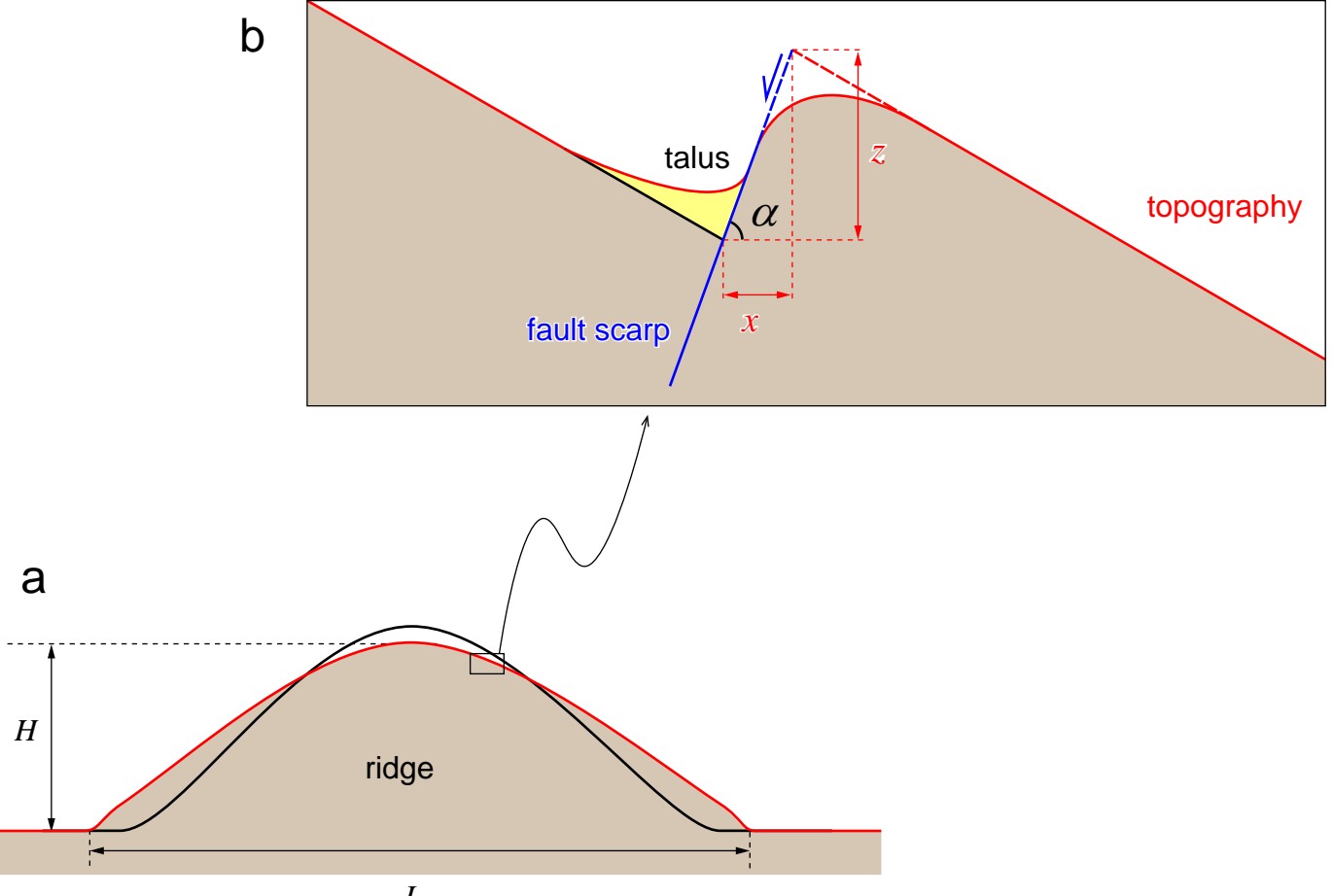

b

talus

topography

fault scarp

$\alpha$

$z$

$x$

a

$H$

ridge

$L$

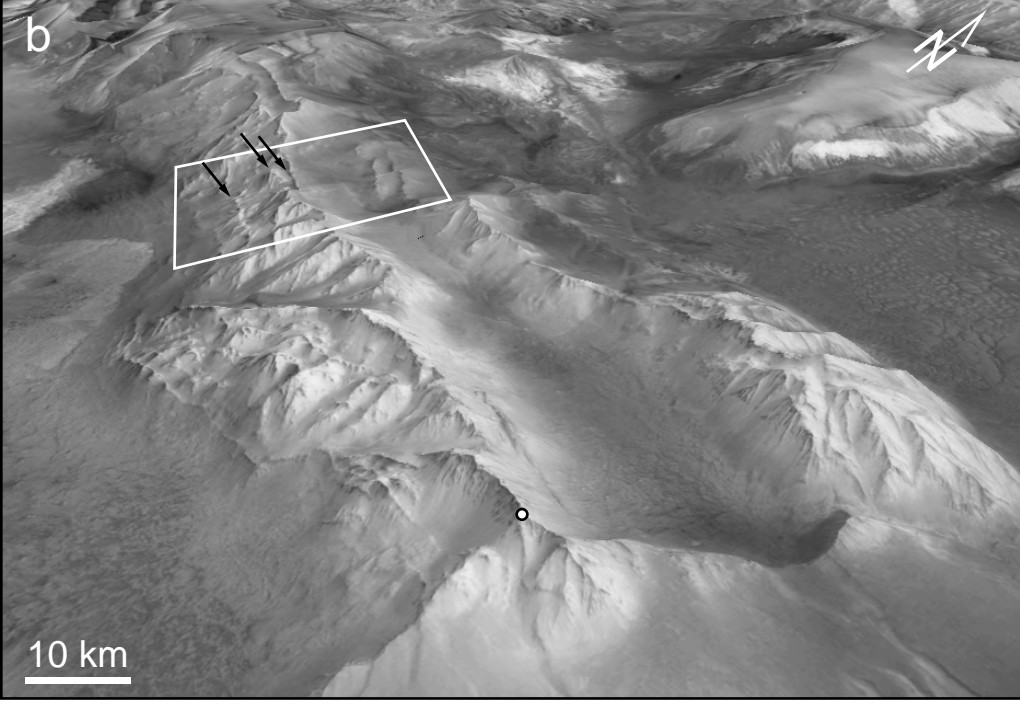

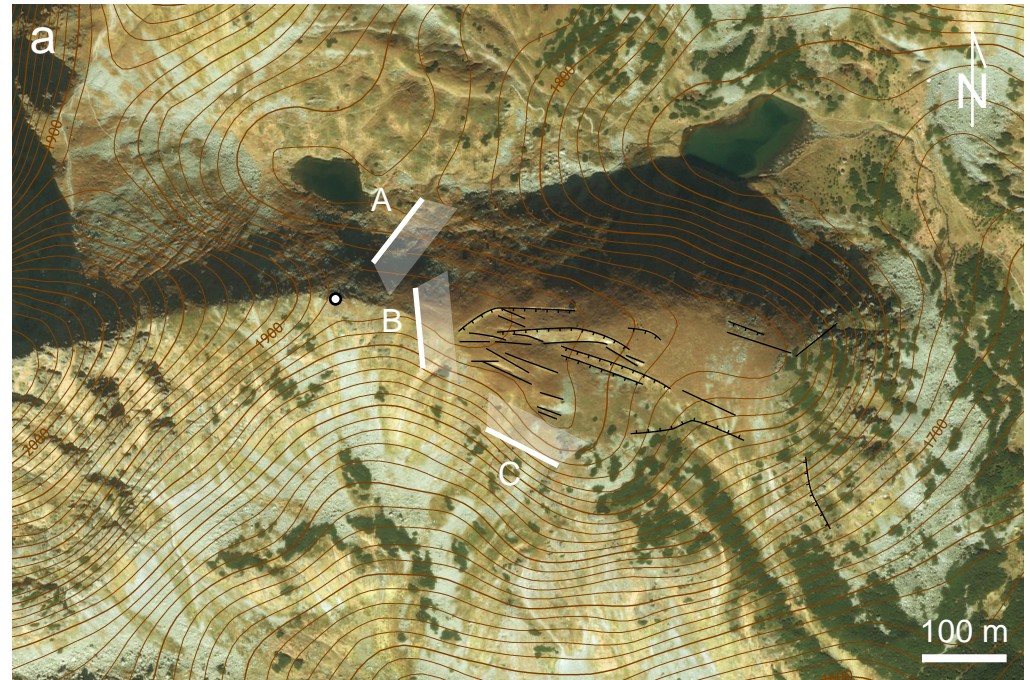

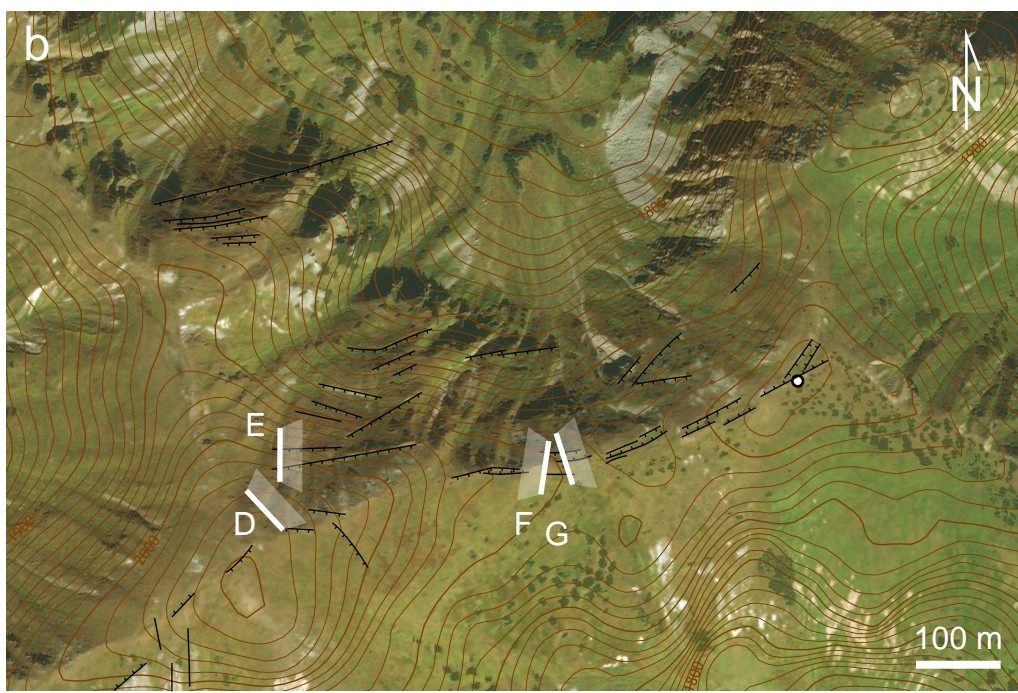

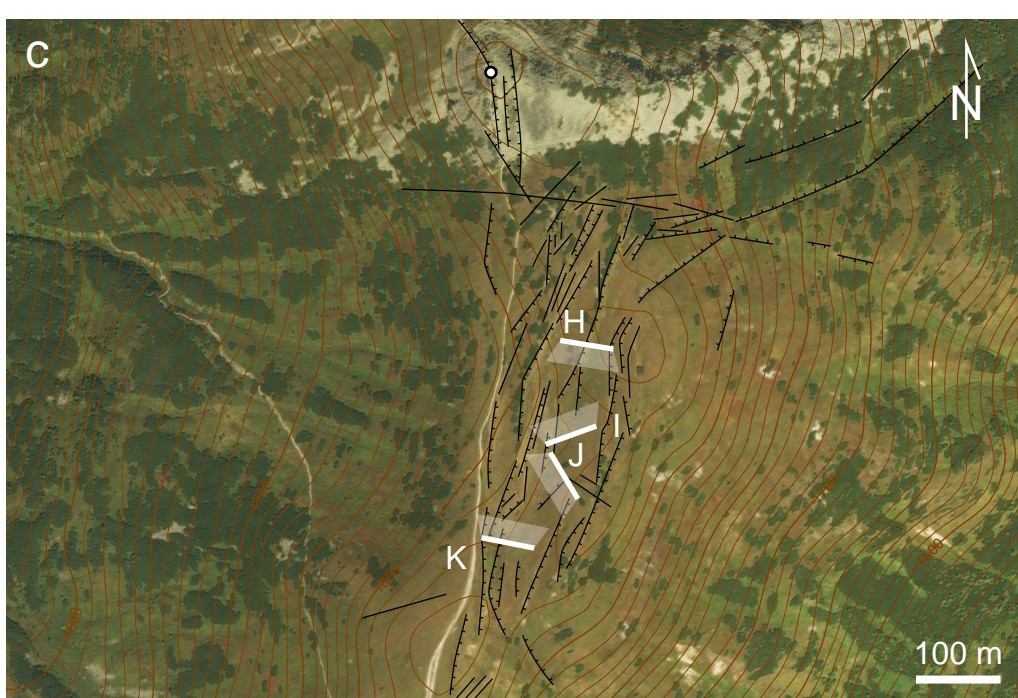

a

A

B

C

b

D

E

F

G

c

H

I

J

K

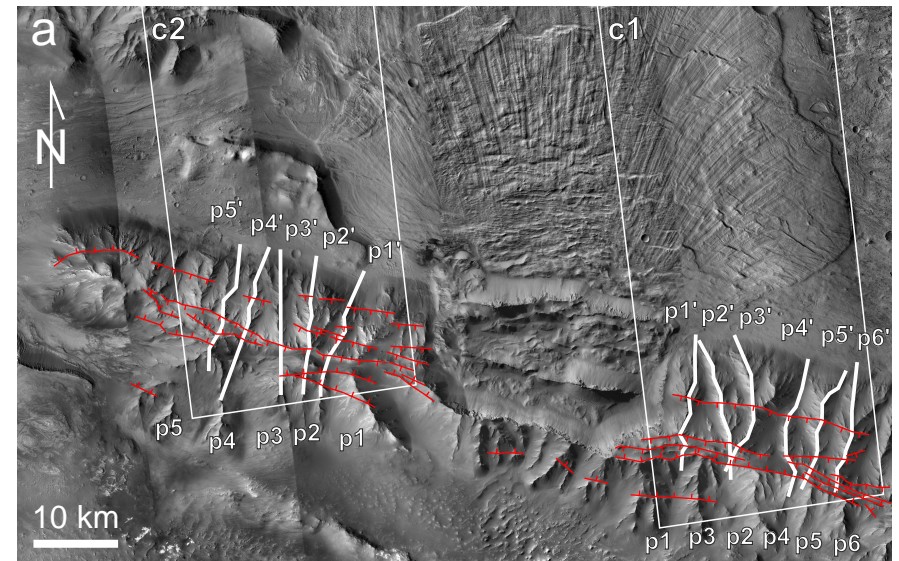

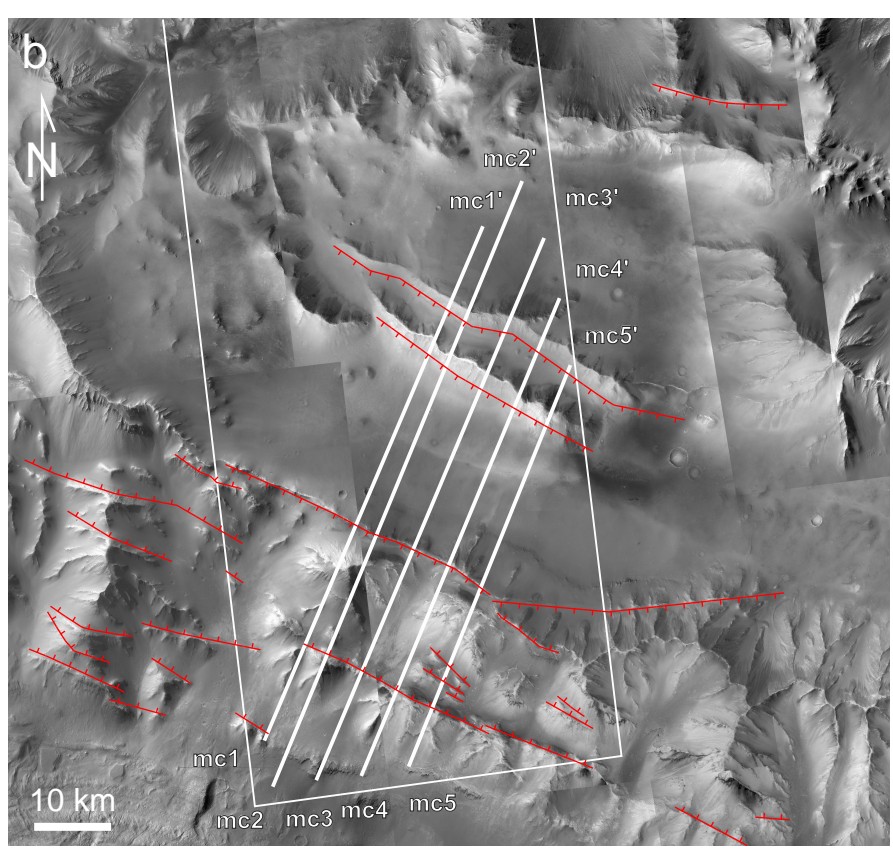

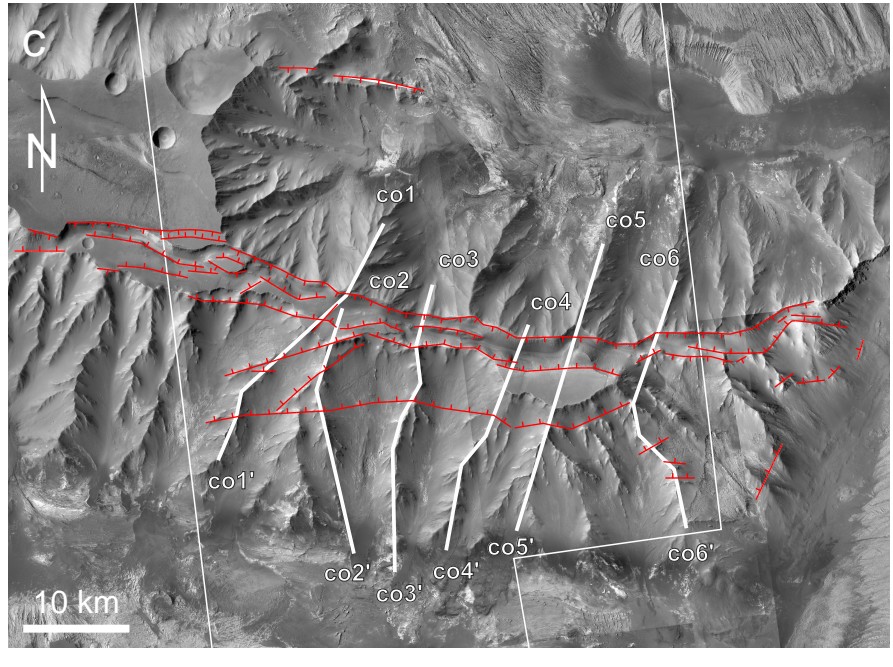

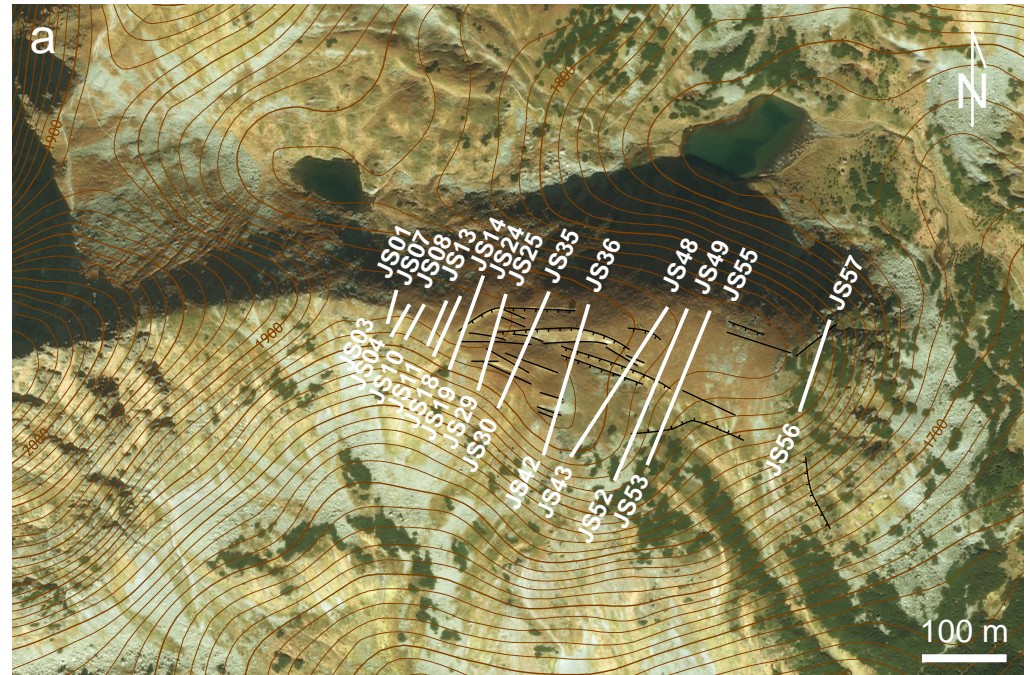

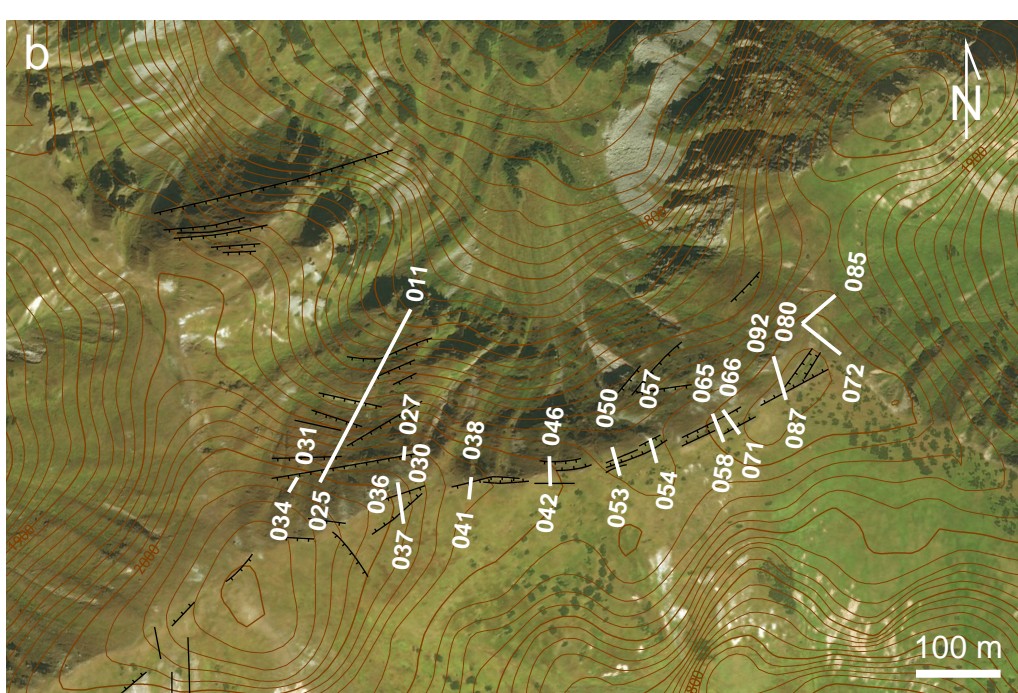

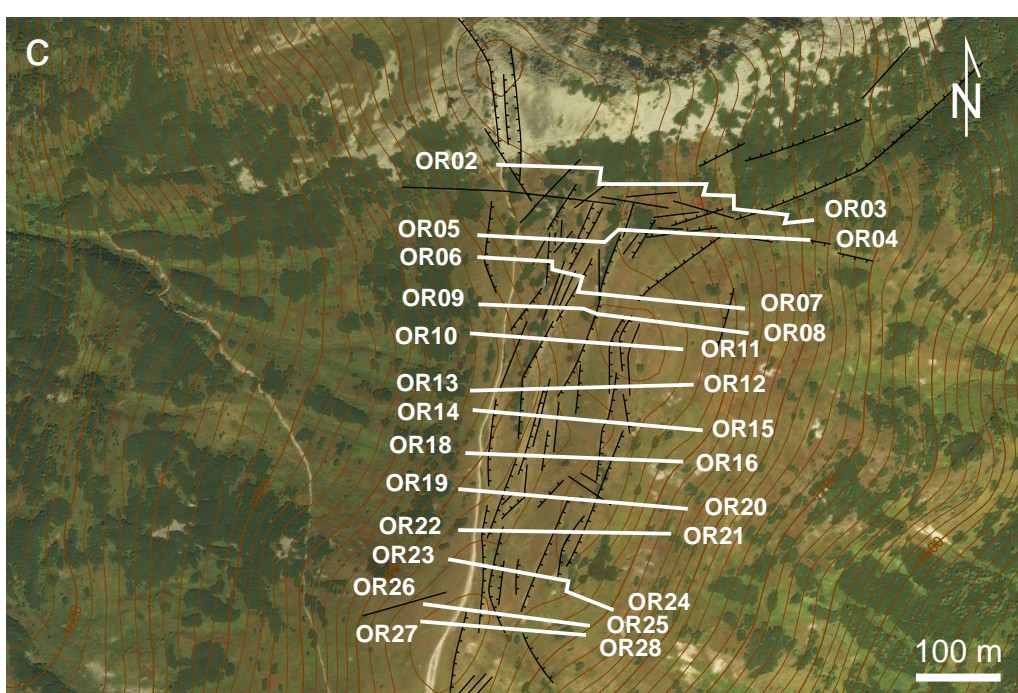

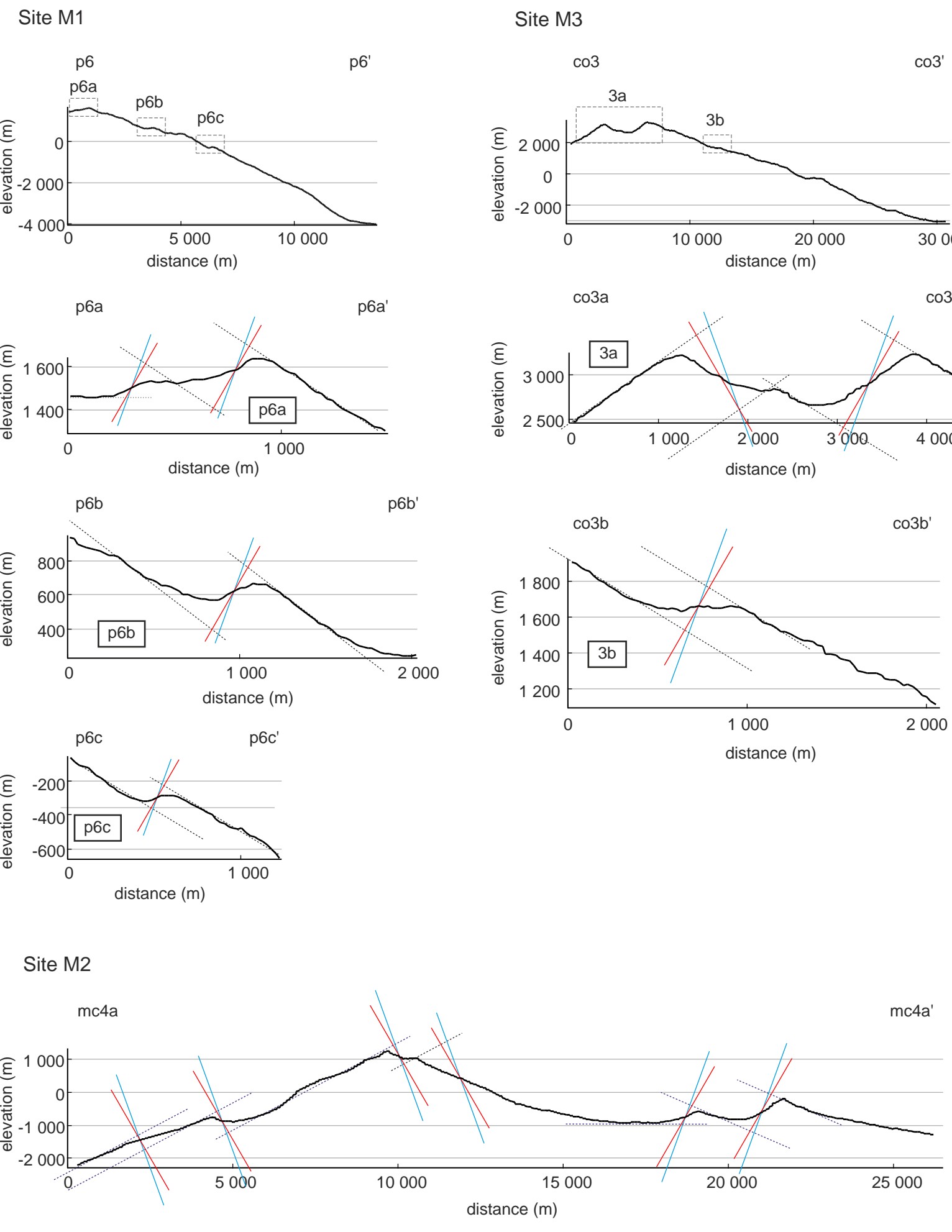

## Site T1

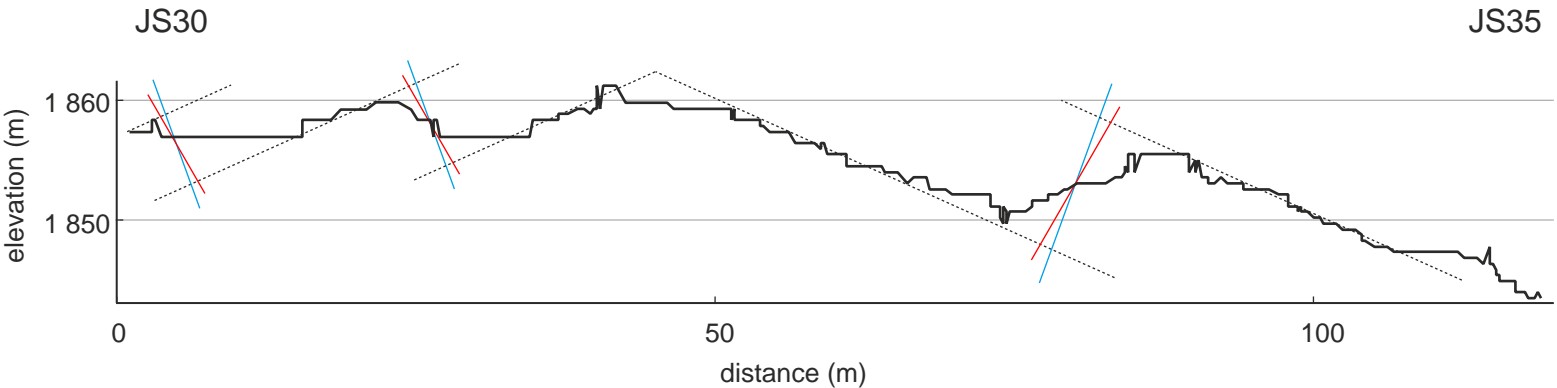

## Site T2

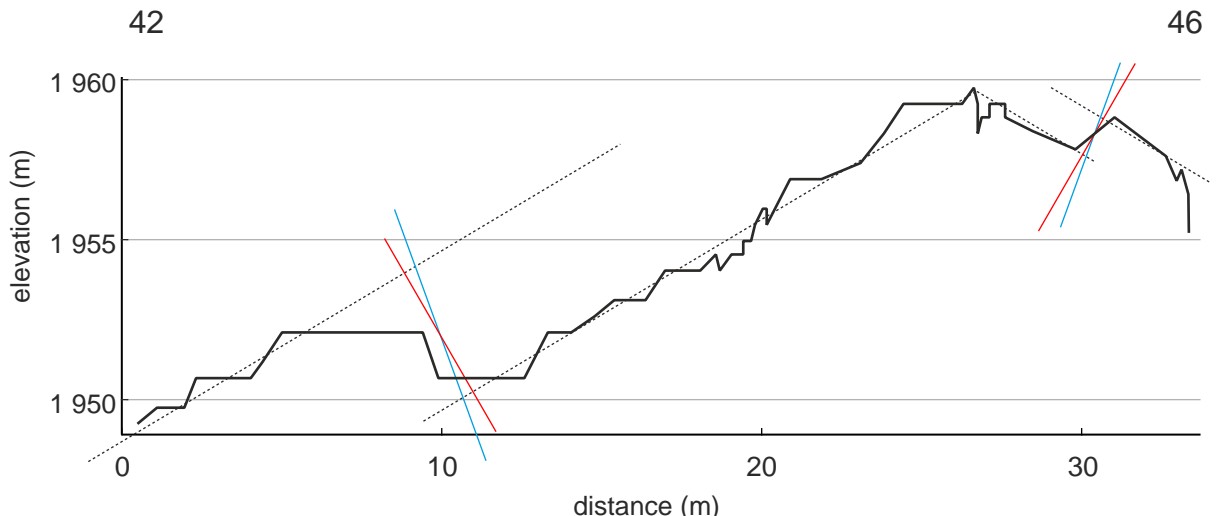

## Site T3

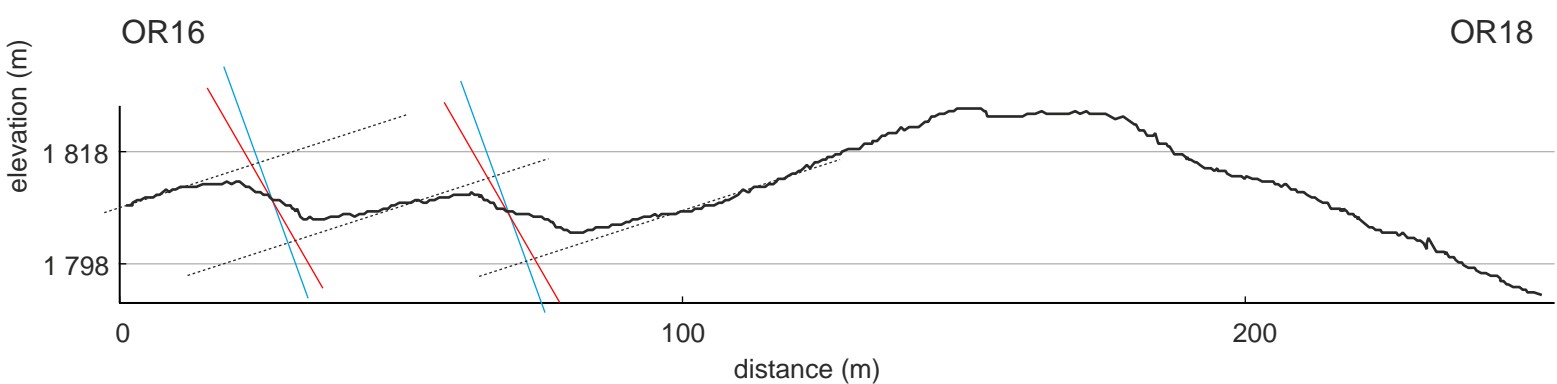

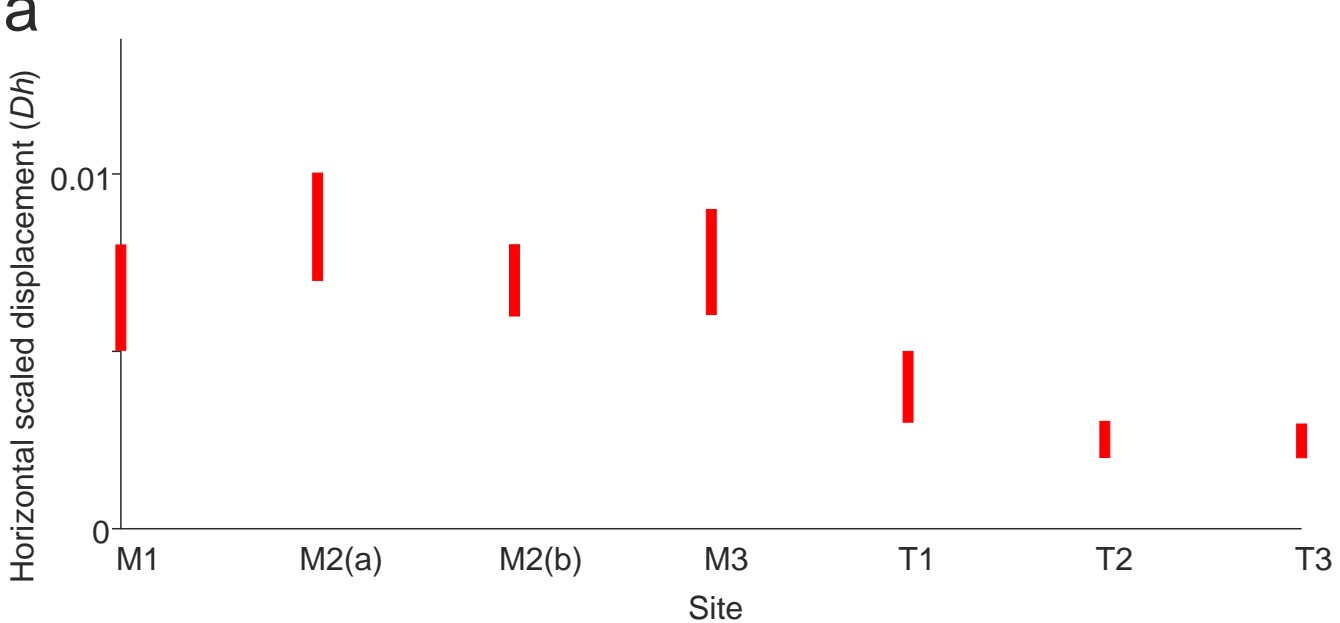

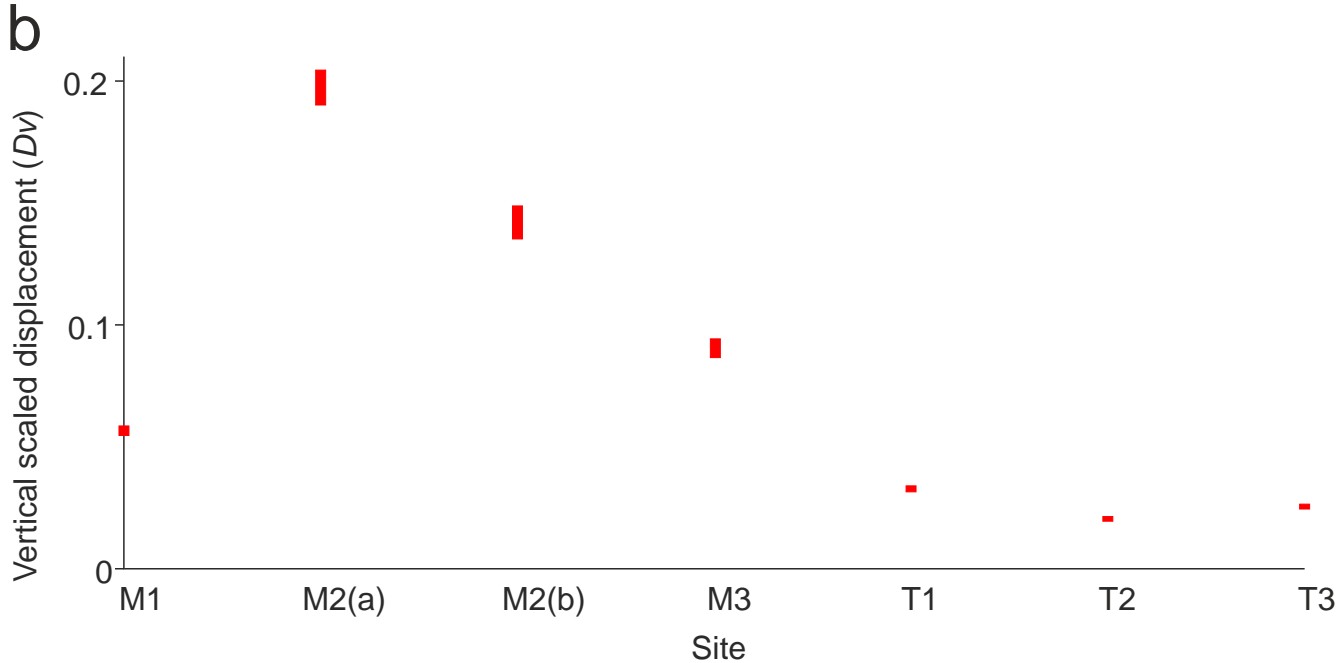

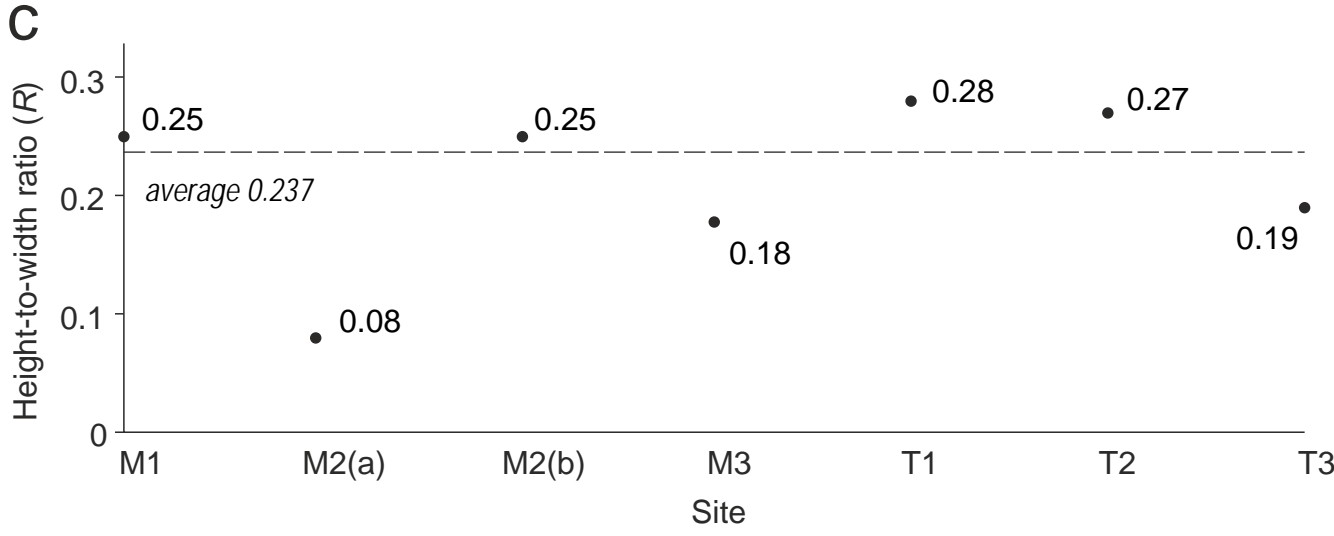