# Peer review of "Deep-seated gravitational slope deformation scaling on Mars and Earth: same fate for different initial conditions and structural evolutions"

_Earth Surface Dynamics, 2018_

## Referee Comment (RC1) · Anonymous Referee #1 · 17 May 2018

GENERAL COMMENTS:

The paper evaluates the evolution of deep-seated gravitational slope deformation (DS-GSD) processes on both Earth (Tatra Mountains) and Mars (Valles Marineris). The approach of analogue studies is very interesting, and the text is generally in a good condition. But there are still some minor improvements I recommend to consider.

SPECIFIC COMMENTS:

**1 (Page 2): I recommend to extend the introduction by a small chapter providing**

definitions and basic information about DSGSD and the motivation of this work. The paper immediately starts with a relatively specific chapter about the hillslopes of Valles Marineris.

**2 (Page 3 Line 14-18 and Page 4 Line 25-28): The hypothesis of a glaciated Valles Marineris is not that common. So it is necessary to provide better citations. Laskar et al., 2004 only provides information about the variations of solar insolation on Mars. It does not mention anything about a glaciated Valles Marineris. Moreover, high obliquity excursions may cause a cold climate; but this does not necessarily mean, that there is enough ice/water causing the development of glaciers. Please cite Mège and Bourgeois, 2011 and Gourronc et al., 2014 already here.**

TECHNICAL CORRECTIONS:

**3 (Page 1 Line 24): "The large offsets make necessary reactivation of the DSGSD fault scarps in Valles Marineris, ...". Do you mean "The large offsets necessarily reactivate the DSGSD fault scarps in Valles Marineris, ..."? Please clarify.**

**4 (Page 2 Line 13): "streams erosion", maybe "fluvial erosion" would be better.**

**5 (Page 2 Line 16): Usually, the maximum depth of Valles Marineris is given with 7 km, not 10 km.**

**6 (Page 2 Line 16): "...whereas typical mountain slopes on Earth are hundreds to a few kilometres high." It should be "... whereas typical mountain slopes on Earth are hundreds of meters to a few kilometers high."**

**7 (Page 3 Line 25): Please add where the Socompa rock avalanche took place.**

**8 (Page 4 Line 12-13): I guess the sentence "The ridge material that makes the lower part of the ridges are usually covered by debris slopes." should be "The ridge material that makes the lower part of the ridges is usually covered by debris slopes.".**

**9 (Page 5 Line 8): "resolultion" into "resolution".**

**10 (Page 10 Line 3 and 9): "A few uphill-facing normal faults scarps" into "A few uphill-facing normal fault scarps".**

**11 (Page 16 Line 15): "Because DSGSD is observed at surface..." into "Because DSGSD is observed at the surface...".**

**12 (Page 18 Line 5-7): It guess "..., which suggests that this conclusion..." should be "..., suggesting that this conclusion...".**

**13 (Page 18 Line 21): Please cite Mège and Bourgeois, 2011 and Gourronc et al., 2014 here too, besides Laskar et al., 2004.**

IMAGES:

**14 (Fig. 1b and 5): It would be nice for a better comparison to present a shaded relief map of the Tatra study area too, instead of a satellite image. In Fig. 5 I would add a shaded DEM too (but keep the satellite imagery here!). Maybe you can put the shaded DEM here on the right side of each related image. And maybe you put the perspective photographs in a separate figure, so they can be presented much larger.**

**15 (Fig. 2): The white box indicating the HiRISE image on the right is not accurately placed. It should be more left. Moreover, it would be better if the images are aligned as CTX photo/CTX DEM/HiRISE.**

**16 (Fig. 2, 4, and 6): I recommend to add the approximate coordinates where each image is centered at. This makes it easy for readers to find these areas themselves. Moreover, you can consider adding the image numbers of the used images too.**

---

## Referee Comment (RC2) · Anonymous Referee #2 · 17 May 2018

GENERAL COMMENTS

The paper proposes a comparative morpho-structural study, investigating possible scaling relationships between parameters of DSGSD on Mars and on Earth.

Comparative and scaling approaches are useful tools that can help constraining martian processes, as well as providing environmental constraints not necessarily known for Earth. Therefore, this work has the potential to offer a valid contribution to better understand landscape evolution of Valles Marineris. However, in my opinion, some is-

sues need to be resolved before publication can be recommended. I recognize 3 major issues regarding (1) the assumptions, (2) the lack of a robust validation of the results, and (3) a dubious interpretation of some results. These are discussed in the Specific Comments and (#) more comments are added.

SPECIFIC COMMENTS

(1) (page 4 line 32) "DSGSD in Valles Marineris and the Tatra Mountains are probably not active anymore" but neither reference nor evidence is provided, therefore sounding as a personal opinion. This is a crucial point since discussions on finite strain and its distribution, from which the conclusions are drawn, are based on this assumption (page 16 lines 29-30 "The ridge aspect ratio R [..] informs on whether DSGSD, which is thought to have stopped on all the ridges, attained a similar final state.").

(2) The authors consider the ridge aspect ratio R as evidence of the slope's state of maturity (as read earlier at page 16 lines 29-30 and also at page 18 lines 1-3 "It may be inferred that on the one hand, the maturity or immaturity (instability) of ridges affected by DSGSD may be inferred from their aspect ratio."). No mention is made about previous studies, neither terrestrial nor martian, that adopt R as a proxy for slope's maturity in DSGSD. If this work is actually first of its kind, a robust validation of the results needs to be made. Otherwise, citations are needed. The authors present 6 cases of inactive DSGSD considered of post-glacial origin, 3 martian and 3 terrestrial. In order to accept the aspect ratio R as evidence of slope maturity, more cases have to be presented. In fact, also the authors mention at Page 18 lines 5-8: "Individual fault displacements across DSGSD scarps in the Tatra Mountains are similar to fault displacement in most DSGSD sites on Earth [. . .], which suggests that this conclusion may be extrapolated to other regions. Nevertheless, similar analyses need to be conducted in other ridges affected by DSGSD, both inactive and active, before general conclusions can be drawn.". I would also include cases of terrestrial DSGSD of no-post-glacial origin. This could have two implications: a) if height-to-width ratios for specific categories are found, then the height-to-width ratio for DSGSD will gain meaning with interesting implications for

**ESurfD**
martian instances; 2) if height-to-width ratios do not match specific categories of DS-GSD, then this scaling relationship cannot be used as diagnostic of maturity of ridges affected by DSGSD.

(3) The third major concern arises around the two possible interpretations of the martian Site 2 and the way the results are interpreted based on these two options: (page 16 line 31) "The range of R is narrow (0.18 – 0.29) for Earth and Mars if the glacial valley at Site 2 is fully erosional [..], as interpreted by Gourronc et al. (2014). R values are much more scattered (0.08 – 0.29) and atypical compared with the two other Martian sites if the central valley in Site 2 is of DSGSD origin only [..]. Because of data coherence, we find the interpretation that the glacial valley is of fully erosional origin more likely.". In my opinion, the word atypical already shows an incorrect bias towards one interpretation rather than the other. Atypical means not representative of a type. However, as I said in point 2, this study is not statistically robust enough to actually define a result atypical. I also find the expression "Because of data coherence" and the interpretation that follows not acceptable. This looks to me selecting certain results while excluding others depending on the assumption they please the most. No results should be discarded on the basis of assumptions. Link to this is the issue that I recognize with the second assumption "DSGSD in Valles Marineris is due to post-glacial stress release on slopes". In my opinion, this is an unnecessary assumption, which rather risks to appear as a bias in the conclusions presented by the authors. Rather, the results should provide further support for the post-glacial origin interpretation. It is clear that the authors favour the interpretation of these landforms given by Mege & Bourgeois (2011) and Gourronc (2014), whose observations I find intriguing, but this cannot sound as a bias. I do not think this work shows a balanced review of the alternative explanations for the landforms observed in Valles Marineris, as instead done by Mege & Bourgeois (2011) and Gourronc (2014) in their introductory paragraphs.

**A value of ∼ 0.24 is given as representative of all the instances of DSGSD. However, it is never said how this value is obtained and, only by going through Table 1, I understood**
that ∼ 0.24 is an average value. I think providing the range of values within which the cases are found is more appropriate. Also, this links to the discussion in point 2 of the major issues.

**In the abstract "On both planets, strain is distributed over the same number (∼5) of major scarps;.." – If the number of the major faults is the same, it cannot be ∼5.**

**In the abstract "The measured finite strain of the Valles Marineris ridges is 3 times larger than in the Tatra Mountains,.." – This ratio is mentioned only at Page 15 line 4 "Dh for the Valles Marineris sites (-0.006) is roughly three times the values for the Tatras sites (-0.002) only.". However, in 2.2 Methods, Dh is called "scaled horizontal displacement" and nowhere else "finite strain" is used. I think more attention should be paid on clearly specify the name of the parameters, so to make it easier to follow the calculations and discussions.**

**The introduction does not sound well structured. A clear general introduction and description of DSGSD, which is the subject of this work, is missing. I would then discuss the evidence of such landforms on Mars. I would mention the importance of comparative planetology studies and scaling approaches, because this is what the work is about. I would conclude by clearly stating the aims of the study, which I struggle to find; explain why you are doing it.**

**Sub-paragraph 1.3. I do not think the authors manage to properly convey the message. First of all, I would refer to volume rather than using expressions such as "Landslides that are small with respect to mountain size" or "For landslides that involve a large fraction of the mountain slope". The effect of volume on landslide mobility is indeed not fully understood and long runout landslides, which the authors clearly refer to, are characterized by volumes bigger than 1 million cubic meters. Important examples of terrestrial long runout landslides other than the Socompa rock avalanche exist, so I would extend the citations (e.g., Saidmarreh l., Blackhawk l., Heart Mountain l., Turtle Mountain l.). To present, many papers have try to contribute to the understanding**

of the influence of different parameters (volume, slope, gravity, fluid content, and so on) on landslide mobility. Just to name a few, McEwen (1989); Soukhovitskaya and Manga (2006); Lucas and Mangeney (2007); Lucas et al (2011); Johnson and Campbell (2017). Certainly this discussion is far from being close. For this reason I think that the sentence at Page 3 lines 29-32 ("This dependency of landslide propagation on slope, and indirectly on volume (and friction), initially identified on laboratory experiments (Farin et al., 2014), could be adequately documented by natural examples thanks to some very large Martian landslides (Borykov et al., submitted), much larger than any terrestrial landslide, which help populate the landslide dataset for voluminous landslides that propagate on nearly flat surfaces."), does not pay the right tribute to the papers that have already been published on long runout landslides and does not correctly report about the current knowledge on the influence of different parameters.

**There is no mention on whether authors made the DEMs or not. If they did, I would mention what software they used (e.g., SocetSet or Ames Stereo Pipeline).**

**Page 3 lines 2-5 "Uphill-facing normal faulting and crestal extensional deformation is indeed well documented on Earth in areas of DSGSD (review in Mège and Bourgeois, 2011). In most described terrestrial instances, such as in the Alps of Europe, Japan and New Zealand, the Andes and others, where DSGSD has been described in mountain ridges glaciated during the Quaternary (review in Mège and Bourgeois, 2011),.." – the use of "others" is not appropriate. Moreover, Mège and Bourgeois (2011) is not a review paper. More relevant papers about terrestrial cases can be cited.**

**Page 3 lines 8-9 "..and additionally provides a good framework to understand the detected mineralogical occurrences as from CRISM (Mège and Bourgeois, 2011; Cull et al., 2014)." – Mège and Bourgeois (2011) do not work with CRISM data but they mention the identification of sulphates and hydrated silica covering the floor of Valles Marineris providing references. Even more recently, Watkins et al. (2015) provide evidence of hydrated silicate using CRISM data.**

**Page 5 lines 6-7 "The observations of the Valles Marineris trough system reported here were done using Mars Reconnaissance Orbiter/CTX imagery as a baseline (Fig. 2)." – I would put a reference to Figure 1a and I would move it soon after Valles Marineris, removing "(Fig.2)".**

**Page 6 lines 2-3 "DEMs generated from CTX stereo pairs have the appropriate vertical precision of ca. 10 m,.." – Vertical precision of CTX stereo-derived DEMs is not so straightforward to assess. It is usually reported as several meter. Also, in Fig. 2, a different value is reported (? 15 m). Commonly in the literature, pixel resolution is given for CTX-DEM ($\sim$20 m).**

**Page 6 line 17 and Page 7 lines 1-2 "the faults cutting the profiles are located and marked as two lines illustrating two mean fault dip angles, $\alpha$=60$^\circ$ and $\alpha$=70$^\circ$ (Fig. 3), representative of unrotated shallow normal faults in extensional settings on Earth (e.g., Gudmundsson, 1992; Acocella et al., 2003). – I could not find any reference to fault dip angles in Acocella et al. (2003). In Gudmundsson (1992) I have found "Fault dip ranges from 42$^\circ$ to 89$^\circ$, with a mean of 73$^\circ$, but nearly 80% of the fault dip between 65$^\circ$ and 79$^\circ$". I am not saying that the range 60$^\circ$ and 70$^\circ$ is not reasonable, I just do not see the justification in the literature proposed. Also, the authors clearly show that they are in favour of a DSGSD with a post-glacial origin scenario, almost discarding alternative explanations (Page 2 lines 20-22 "They denote extensional tectonics, but boundary forces that result in crustal "rifting" are unlikely to be the cause of this deformation even though rifting is frequently considered to have been a major contributor to the formation of some of the main Valles Marineris chasmata"). For this reason, I find slightly incoherent the reference to Gudmundsson (1992) and Acocella et al. (2003), who worked in the rift zone of Iceland and on flank instability of Mount Etna, respectively. Is there any fault dip range given for DSGSD in the literature?**

**Page 7 line 1 "..and marked as two lines illustrating two mean fault dip angles, $\alpha$=60$^\circ$ and $\alpha$=70$^\circ$ (Fig. 3).." – Figure 3 show just one line representing a general fault. So, I would remove the reference to Figure 3 at this line.**

**Page 7 lines 6-7 "The 10° angle interval is considered as the error on the true angular value, which cannot be retrieved from topography.." – I would place this sentence near to when fault dip range is discussed. This is the explanation to why the range 60°-70° is considered.**

**Page 8 line 9 "Three sites [..] were selected in Valles Marineris, based on DEM generation possibility,.." – Without DEM the study cannot be conducted, so not necessary.**

**Page 10 lines 5-9 "Site 5 [..] is a ridge west of VeÄ¿ká Garajova Kopa on the way to VeÄ¿ká Kopa. [..] The 900 m wide ridge rises up to 250 m above the adjacent valleys. The height to width ratio R is 0.28." – These values correspond to Site 4 in Table 1 (??)**

**Page 11 line 3 "topographic profiles were measured" –Profiles are traced. See also caption Fig.4.**

**Page 15 line 4 "Dh for the Valles Marineris sites ($\sim$0.006) is roughly three times the values for the Tatras sites ($\sim$0.002) only." – I did the calculation for Dh and I cannot obtain the same values. For terrestrial sites (4, 5, 6) I get $\sim$ 0.003, whereas for martian sites I get $\sim$0.008 if sites 1, 2b, 3 are taken; $\sim$0.008 if sites 1, 2a, 2b, 3 are taken; $\sim$0.007 if sites 1, 2a, 3 are taken; $\sim$0.006 if sites 1, 3 are taken. Please, explain how you did the calculation.**

**Page 16 lines 2-3 "This is interpreted as a consequence of removal of the highest part of the ridge, as discussed in Section 5." – I cannot find the discussion in Section 5.**

**Page 16 lines 11-12 "..the number of DSGSD faults in Valles Marineris (Tables 2 to 4) is not significantly different from the number of such faults in the Tatra Mountains." – No count of faults is given in any table. Also, there is not a Table 4.**

**Page 16 line 13 "The very large fault offsets measured on individual faults in Valles Marineris require cumulated events (e.g., Fossen, 2010)." – Fossen (2010) is a textbook. If this has to be the reference, please provide the chapter and the page where**

the information can be found.

COMMENTS ON FIGURES:

**Figure 2 – I think this is a superfluous figure.**

**Figure 3 – I would add a sketch/model showing which are the reference points from where the height and the width of a ridge are measured. For example, this information is always present in papers on long runout landslides, making clear that height drop and length of a landslide are measured from the highest point of the scarp to the furthest point of the deposit.**

**Figures 4, 5, 6, 7 – I would create a border/space between the images.**

**Figure 6 – I think in 6b and 6c codes of the area ("mc" and "co", respectively), then used in the name of the profiles in Fig.8a, are missing. Also, it took me a bit to find the codes "c1" and "c2" in 6a. I would create a box around the profiles.**

**Figure 8a, 8b – I think unit of measure should be placed on the graphs rather than in the caption.**

COMMENTS ON TABLES:

**I would not use progressive numbers for site IDs, rather some other way that would help to identify immediately on the graphs which case is from which planet.**

**Table 2 – I suggest using the symbol x bar and z bar instead of writing "Mean horizontal fault displacement" and "Mean vertical displacement", respectively.**

**Table 2 – I suggest it to have the same style of Table 3. In Table 2, $\alpha$ is placed under Site ID, which does not make sense.**

**I would group Table 2 and 3 (and maybe also 1).**

**Values of Dh and Dv are not given in any table.**

COMMENTS ON SUPPLEMENTARY MATERIAL:

\# I think that the data that the graphs are trying to show should be better made available in a table. On the graphs, the values of the displacement is not clear, making it difficult to follow the calculations that have brought to the results reported in the manuscript.

\# I would add all the profiles obtained for this study, rather than just having few examples in the manuscript.

---

## Author Response (AR1)

Referee's comments are in black.

Response is in blue.

Manuscript changes are in green.

GENERAL COMMENTS:

The paper evaluates the evolution of deep-seated gravitational slope deformation (DSGSD) processes on both Earth (Tatra Mountains) and Mars (Valles Marineris). The approach of analogue studies is very interesting, and the text is generally in a good condition. But there are still some minor improvements I recommend to consider.

SPECIFIC COMMENTS:

**1 (Page 2): I recommend to extend the introduction by a small chapter providing definitions and basic information about DSGSD and the motivation of this work. The paper immediately starts with a relatively specific chapter about the hillslopes of Valles Marineris.**

The revised manuscript includes a new section inserted at the beginning of the introduction that follows this recommendation:

"Deep-seated gravitational slope deformation (DSGSD) is a slow process that deforms mountain ridges, usually as a paraglacial process (e.g., Mège and Bourgeois, 2011) that is readily identified by uphill-facing normal fault scarps in the upper part of ridge slopes, resulting in ridge-top splitting and summital valley development. Extension at the upper part of the ridge, in the absence of catastrophic slope failure, results in many instances in compressive bulging (e.g., Radbruch-Hall et al., 1976; Beget, 1985; Savage and Varnes, 1987; Chigira, 1992; Reitner et al., 1993; Ambrosi and Crosta, 2006; Hippolyte et al., 2006; Discenza et al., 2011). In some cases, the slope is observed to be overthrusting the valley (Mahr, 1977; Guerricchio and Melidoro, 1979; Bachman et al., 2009; Savage and Varnes, 1987), which suggests that a décollement may connect the faults in the upper part of the slope with displacement at the bottom of the slope. Such décollements are more frequently inferred than observed in the field, and in some cases there is evidence that fault slip in the upper part of the slope is not connected by a continuous plane to the lower slope, and displacement in the ridge may be more diffuse, for instance due to the presence of shales (Radbruch-Hall et al., 1976; Beget, 1985; Hürlimann et al., 2006). In some instances, the bottom of the slope is not deformed and the extension in the upper part of the ridge is absorbed within the ridge, the density of which is locally increased (Discenza et al., 2011), possibly by crushing (Beck, 1968; Mahr, 1977). Makowska et al. (2016) found that in homogeneous rock indeed, DSGSD in the upper part of the slope and at the bottom are disconnected, and that an internal core of intensely deformed rock underlies the uphill-facing scarps developed in the upper part of the slope.

Such deformation has been reported on bedrock hillslopes in Valles Marineris on Mars (Mège and Bourgeois, 2011), but at the scale that is much larger than on Earth. For instance, ridge-top splitting on Earth generates scarps that rarely exceed metres in height, whereas on Mars, a quick look at digital elevation models shows that scarps are tens to hundreds of metres are not rare. This work quantifies this intriguing difference and explores some implications."

**2 (Page 3 Line 14-18 and Page 4 Line 25-28): The hypothesis of a glaciated Valles Marineris is not that common. So it is necessary to provide better citations. Laskar et al., 2004 only provides information about the variations of solar insolation on Mars. It does not mention anything about a glaciated Valles Marineris. Moreover, high obliquity excursions may cause a cold climate; but this does not necessarily mean, that there is enough ice/water causing the development of glaciers. Please cite Mège and Bourgeois, 2011 and Gourronc et al., 2014 already here.**

Done

TECHNICAL CORRECTIONS:

**3 (Page 1 Line 24): "The large offsets make necessary reactivation of the DSGSD fault scarps in Valles Marineris,...". Do you mean "The large offsets necessarily reactivate the DSGSD fault scarps in Valles Marineris,..."? Please clarify.**

The authors mean that having such large offsets implies that the faults are reactivated because a single slip cannot generate such a large offset.

In the revised manuscript we use the formulation "imply reactivation", which is probably more clear than the initial wording.

**4 (Page 2 Line 13): "streams erosion", maybe "fluvial erosion" would be better.**

Yes indeed. Modified.

**5 (Page 2 Line 16): Usually, the maximum depth of Valles Marineris is given with 7 km, not 10 km.**

After checking the HRSC digital elevation model of Valles Marineris, we find that 8 km is not rare locally the depth attains 9 km or a few hundred metres more locally in Ius, Coprates and Candor chasmata.

The value of 10 km is therefore changed to 9 km in the revised manuscript.

**6 (Page 2 Line 16): "...whereas typical mountain slopes on Earth are hundreds to a few kilometres high." It should be "...whereas typical mountain slopes on Earth are hundreds of meters to a few kilometers high."**

Modified.

**7 (Page 3 Line 25): Please add where the Socompa rock avalanche took place.**

Chile is added.

**8 (Page 4 Line 12-13): I guess the sentence "The ridge material that makes the lower part of the ridges are usually covered by debris slopes." should be "The ridge material that makes the lower part of the ridges is usually covered by debris slopes."**

Thanks for noting this. Changed.

**9 (Page 5 Line 8): "resolultion" into "resolution".**

Same - changed.

**10 (Page 10 Line 3 and 9): "A few uphill-facing normal faults scarps" into "A few uphill-facing normal fault scarps".**

Same - changed.

**11 (Page 16 Line 15): "Because DSGSD is observed at surface ..." into "Because DSGSD is observed at the surface...".**

Same - changed.

**12 (Page 18 Line 5-7): It guess "..., which suggests that this conclusion..." should be"..., suggesting that this conclusion...".**

Same - changed.

**13 (Page 18 Line 21): Please cite Mège and Bourgeois, 2011 and Gourronc et al., 2014 here too, besides Laskar et al., 2004.**

Done

IMAGES:

**14 (Fig. 1b and 5): It would be nice for a better comparison to present a shaded relief map of the Tatra study area too, instead of a satellite image.**

For Figure 1: agreed, done.

In Fig. 5 I would add a shaded DEM too (but keep the satellite imagery here!). Maybe you can put the shaded DEM here on the right side of each related image. And maybe you put the perspective photographs in a separate figure, so they can be presented much larger.

The resolution of available DTMs would make a very pixelated shaded relief image. We prefer to add contour lines to the satellite image (the scarps are however in general too small to be resolved). Note also that the satellite images were changed for images of much better quality. The satellite image + contour is now Figure 4, and the photographs are Figure 5.

**15 (Fig. 2): The white box indicating the HiRISE image on the right is not accurately placed. It should be more left. Moreover, it would be better if the images are aligned as CTX photo/CTX DEM/HiRISE.**

Figure 2 was removed, in response to a comment from Reviewer 2.

**16 (Fig. 2, 4, and 6): I recommend to add the approximate coordinates where each image is centered at. This makes it easy for readers to find these areas themselves. Moreover, you can consider adding the image numbers of the used images too.**

Figure 2 was removed, as requested by Referee 2. One landmark is reported on each of the satellite images of the three Martian and terrestrial sites and its coordinates provided in the caption. This was done for the new Figure 3 and Figure 4 (Landmarks were not added to Figures 6 and 7, which display the same satellite images as Figures 3 and 4).

Referee's comments are in black.

Response is in blue.

Manuscript changes are in green.

Page and line numbers refer to the submitted manuscript, except otherwise indicated.

GENERAL COMMENTS

The paper proposes a comparative morpho-structural study, investigating possible scaling relationships between parameters of DSGSD on Mars and on Earth.

Comparative and scaling approaches are useful tools that can help constraining martian processes, as well as providing environmental constraints not necessarily known for Earth. Therefore, this work has the potential to offer a valid contribution to better understand landscape evolution of Valles Marineris. However, in my opinion, some issues need to be resolved before publication can be recommended. I recognize 3 major issues regarding (1) the assumptions, (2) the lack of a robust validation of the results, and (3) a dubious interpretation of some results. These are discussed in the Specific Comments and (#) more comments are added.

This is a very thorough review that goes into many aspects of the work. The authors are happy to provide clarification on all the points addressed in this review, which will certainly substantially improve the manuscript.

SPECIFIC COMMENTS

(1) (page 4 line 32) "DSGSD in Valles Marineris and the Tatra Mountains are probably not active anymore" but neither reference nor evidence is provided, therefore sounding as a personal opinion. This is a crucial point since discussions on finite strain and its distribution, from which the conclusions are drawn, are based on this assumption (page 16 lines 29-30 "The ridge aspect ratio R [..] informs on whether DSGSD, which is thought to have stopped on all the ridges, attained a similar final state.").

This is indeed critical. This sentence is updated in the revised manuscript:

"DSGSD in Valles Marineris and the Tatra Mountains are probably not active anymore, being dated Hesperian to Lower Amazonian for Valles Marineris (Mège and Bourgeois, 2011) and before the Late Quaternary for the Tatras (Panek et al., 2017),…"

(2) The authors consider the ridge aspect ratio R as evidence of the slope's state of maturity (as read earlier at page 16 lines 29-30 and also at page 18 lines 1-3 "It may be inferred that on the one hand, the maturity or immaturity (instability) of ridges affected by DSGSD may be inferred from their aspect ratio."). No mention is made about previous studies, neither terrestrial nor martian, that adopt R as a proxy for slope's maturity in DSGSD. If this work is actually first of its kind, a robust validation of the results needs to be made. Otherwise, citations are needed. The authors present 6 cases of inactive DSGSD considered of post-glacial origin, 3 martian and 3 terrestrial. In order to accept the aspect ratio R as evidence of slope maturity, more cases have to be presented. In fact, also the authors mention at Page 18 lines 5-8: "Individual fault displacements across DSGSD scarps in the Tatra Mountains are similar to fault displacement in most DSGSD sites on Earth [. . .], which suggests that this conclusion may be extrapolated to other regions. Nevertheless, similar analyses need to be conducted in other ridges affected by DSGSD, both inactive and active, before general conclusions can be drawn.". I would also include cases of terrestrial DSGSD of no-post-glacial origin. This could have two implications: a) if height-to-width ratios for specific categories are found, then the height-to-width ratio for DSGSD will gain meaning with interesting implications for martian instances; 2) if height-to-width ratios do not match specific categories of DSGSD, then this scaling relationship cannot be used as diagnostic of maturity of ridges affected by DSGSD.

As far as we know, no "maturity" or "stability" criterion has been defined by other authors earlier. The "maturity" of ridges suggested by the *R* parameter simply reflects the observation that all the ridges, which are now inactive (which is now better ascertained in the manuscript thanks to the previous comment), have all the same *R*, although starting from a different rige geometry.

This cannot be validated other than from the measurements reported here; realistic experimental models and numerical simulations would be required, which would make another paper. For this reason, we now "propose" (Sections 5.2. + 5.3) then "suggest" (Conclusion) that *R* is a maturity parameter that needs to be confirmed by further studies (In the previous version of the manuscript, we "inferred" in the conclusion that R may be such a parameter). The conclusion was then that more analyses are required to confirm this statement before it can be generalized; we agree with the referee that ridges in non-postglacial situations should also be considered in lines 7-8 of page 18 and insert this note. Taking this important comment into account, the manuscript has been improved and now reads like this:

> "The ridge aspect ratio R (eq. 3) is constant for all or most of the ridges (depending in which situation, 2a or 2b, is the Melas-Candor ridge). Considering that all the ridges are now inactive, it follows that in spite of the different scale, they attained a similar final stage, making R an estimate of DSGSD maturity de facto, with R=0.24 for a mature ridge." (Section 5.2)

> "The proportionally larger deformation in Valles Marineris than in the Tatras, in spite of similar ridge aspect ratio, suggests that DSGSD evolved to its finite stage from an initially different aspect ratio. The collected measurements therefore supports the proposition made in Section 5.2 that the R parameter may measure ridge maturity." (Section 5.3)

> "It may be suggested that on the one hand, the maturity or immaturity (instability) of ridges affected by DSGSD may be inferred from their aspect ratio. […] similar analyses need to be conducted in other ridges affected by DSGSD, formed in post-glacial as well as non-postglacial conditions, both inactive and active, before general conclusions can be drawn." (Conclusion)

instead of:

> "The ridge aspect ratio R (eq. 3) informs on whether DSGSD, which is thought to have stopped on all the ridges, attained a similar final stage." (Section 5.2)

> "The proportionally larger deformation in Valles Marineris than in the Tatras, in spite of similar ridge aspect ratio, suggests that DSGSD evolved to its finite stage from an initially different aspect ratio." (Section 5.3)

> "It may be inferred that on the one hand, the maturity or immaturity (instability) of ridges affected by DSGSD may be inferred from their aspect ratio. […] similar analyses need to be conducted in other ridges affected by DSGSD, both inactive and active, before general conclusions can be drawn." (Conclusion)

(3) The third major concern arises around the two possible interpretations of the martian Site 2 and the way the results are interpreted based on these two options: (page 16 line 31) "The range of R is narrow (0.18 – 0.29) for Earth and Mars if the glacial valley at Site 2 is fully erosional [..], as interpreted by Gourronc et al. (2014). R values are much more scattered (0.08 – 0.29) and atypical compared with the two other Martian sites if the central valley in Site 2 is of DSGSD origin only [..]. Because of data coherence, we find the interpretation that the glacial valley is of fully erosional origin more likely.". In my opinion, the word atypical already shows an incorrect bias towards one interpretation rather than the other. Atypical means not representative of a type. However, as I said in point 2, this study is not statistically robust enough to actually define a result atypical. I also find the expression "Because of data coherence" and the interpretation that follows not acceptable. This looks to me selecting certain results while excluding others depending on the assumption they please the most. No results should be discarded on the basis of assumptions. Link to this is the issue that I recognize with the second assumption "DSGSD in Valles Marineris is due to post-glacial stress release on slopes". In my opinion, this is an unnecessary assumption, which rather risks to appear as a bias in the conclusions presented by the authors. Rather, the results should provide further support for the post-glacial origin interpretation. It is clear that the authors favour the interpretation of these landforms given by Mege & Bourgeois (2011)

and Gourronc (2014), whose observations I find intriguing, but this cannot sound as a bias. I do not think this work shows a balanced review of the alternative explanations for the landforms observed in Valles Marineris, as instead done by Mège & Bourgeois (2011) and Gourronc (2014) in their introductory paragraphs.

This is again an excellent (series of) point(s). This actually is a long story that we estimated not necessary to narrate, but we understand that the manuscript as submitted was not satisfactory, and provide now more details.

Geomorphologic analysis of this area clearly shows that the northern side of the ridge has been eroded by processes which are not found in the other areas studied here. Mège and Bourgeois (2011) found this curious but did not really insist on that example of DSGSD because they did not know really what to do with this. It was taken as an unusual DSGD case. The work reported by Gourronc et al. (2014) was elaborated by the same group after 3 years of additional work and the conclusion was that a totally different mechanism of erosion of the northern Melas-Candor ridge wall needs to be found. Within the framework of a regional glacial landsystem, the evidence of which is widespread in a number of chasmata (and further evidence was later provided by Mège and Gurgurewicz, Geol. Sinica-Eng 90, doi:10.1111/1755-6724.12960), the northern side of the Melas-Candor ridge would naturally form by glacial erosion proceeding from a narrow graben located west of the ridge crest. The graben would widen and deepen, eventually resulting in a structurally controlled U-shaped valley. No other origin has been proposed for this feature as far as we know. Mège and Bourgeois (2001) had therefore hypothesized (2a) whereas Gourronc et al. (2014) corrected this view by proposing (2b). We therefore use interpretation (2a) as a possibility, but our own reasoning now favours our more recent interpretation (2b).

To make this clear, this part of Section 5.2. was in large part rewritten, which gives:

"R values are much more scattered (0.08 – 0.29) and atypical unusual compared with the two other Martian sites if the central valley in Site 2 is of DSGSD origin only (Fig. 9c, Site 2a), as interpreted by Mège and Bourgeois (2011). Here we favour Gourronc et al. (2014), which results in a nearly constant value of R for all the studies ridges, for the following reason.

Geomorphologic analysis of the Candor-Melas ridge shows that the erosional processes that shaped the northern side of the ridge are distinct for the processes that generates the spur-and-gully morphology on the southern side. Mège and Bourgeois (2011) found this unusual but did not succeed in interpreting it further than an advanced stage of DSGSD. The northern side of the ridge was taken as an unusual example of extreme development of DSGSD. Three years after, the same group reinterpreted the chasmata around this feature as part of a huge glacial landsystem, based on many and widespread geomorphological observations (Gourronc et al., 2014). Within the framework of a regional glacial landsystem, the northern side of the Melas-Candor ridge would naturally form by glacial erosion proceeding eastwards from a narrow graben located west of the ridge crest. The graben would widen and deepen, eventually resulting in a structurally controlled U-shaped valley and abrasion and the ridge crest. Mege and Bourgeois (2001) had therefore hypothesized (2a) whereas Gourronc et al. (2014) corrected this view by proposing (2b). Interpretation (2a) is therefore still considered, although we now favour our more recent interpretation (2b)."

instead of:

"R values are much more scattered (0.08 – 0.29) and atypical compared with the two other Martian sites if the central valley in Site 2 is of DSGSD origin only (Fig. 9c, Site 2a). Because of data coherence, we find the interpretation that the glacial valley is of fully erosional origin more likely."

**A value of ~0.24 is given as representative of all the instances of DSGSD. However, it is never said how this value is obtained and, only by going through Table 1, I understood that ~0.24 is an average value. I think providing the range of values within which the cases are found is more appropriate. Also, this links to the discussion in point 2 of the major issues.**

This value is from Figure 9C, which indeed was not mentioned there.

The value of each dot on Figure 9C is now written next to each dot, and this figure is referred to in this part of

the discussion.

**In the abstract "On both planets, strain is distributed over the same number (~5) of major scarps;.." – If the number of the major faults is the same, it cannot be ~5.**

Changed to "strain is distributed over a small number (~5) of major scarps"

**In the abstract "The measured finite strain of the Valles Marineris ridges is 3 times larger than in the Tatra Mountains,.." – This ratio is mentioned only at Page 15 line 4 "Dh for the Valles Marineris sites (-0.006) is roughly three times the values for the Tatras sites (-0.002) only.". However, in 2.2 Methods, Dh is called "scaled horizontal displacement" and nowhere else "finite strain" is used. I think more attention should be paid on clearly specify the name of the parameters, so to make it easier to follow the calculations and discussions.**

"Finite strain" is removed in the abstract, which nows includes the sentence:

"The measured horizontal spreading perpendicular to the ridges is proportionally 3 times larger for the Valles Marineris ridges than the Tatra Mountains".

**The introduction does not sound well structured. A clear general introduction and description of DSGSD, which is the subject of this work, is missing. I would then discuss the evidence of such landforms on Mars. I would mention the importance of comparative planetology studies and scaling approaches, because this is what the work is about. I would conclude by clearly stating the aims of the study, which I struggle to find; explain why you are doing it.**

This echoes a request from Referee 1, which we followed by adding a new introduction section on DSGSD (Section 1.1), wich ends with a clearer presentation of objectives:

"Deep-seated gravitational slope deformation (DSGSD) is a slow process that deforms mountain ridges, usually as a paraglacial process (e.g., Mège and Bourgeois, 2011) that is readily identified by uphill-facing normal fault scarps in the upper part of ridge slopes, resulting in ridge-top splitting and summital valley development. Extension at the upper part of the ridge, in the absence of catastrophic slope failure, results in many instances in compressive bulging (e.g., Radbruch-Hall et al., 1976; Beget, 1985; Savage and Varnes, 1987; Chigira, 1992; Reitner et al., 1993; Ambrosi and Crosta, 2006; Hippolyte et al., 2006; Discenza et al., 2011). In some cases, the slope is observed to be overthrusting the valley (Mahr, 1977; Guerricchio and Melidoro, 1979; Bachman et al., 2009; Savage and Varnes, 1987), which suggests that a décollement may connect the faults in the upper part of the slope with displacement at the bottom of the slope. Such décollements are more frequently inferred than observed in the field, and in some cases there is evidence that fault slip in the upper part of the slope is not connected by a continuous plane to the lower slope, and displacement in the ridge may be more diffuse, for instance due to the presence of shales (Radbruch-Hall et al., 1976; Beget, 1985; Hürlimann et al., 2006). In some instances, the bottom of the slope is not deformed and the extension in the upper part of the ridge is absorbed within the ridge, the density of which is locally increased (Discenza et al., 2011), possibly by crushing (Beck, 1968; Mahr, 1977). Makowska et al. (2016) found that in homogeneous rock indeed, DSGSD in the upper part of the slope and at the bottom are disconnected, and that an internal core of intensely deformed rock underlies the uphill-facing scarps developed in the upper part of the slope.

Such deformation has been reported on bedrock hillslopes in Valles Marineris on Mars (Mège and Bourgeois, 2011), but at the scale that is much larger than on Earth. For instance, ridge-top splitting on Earth generates scarps that rarely exceed metres in height, whereas on Mars, a quick look at digital elevation models shows that scarps are tens to hundreds of metres are not rare. This work quantifies this intriguing difference and explores some implications."

**Sub-paragraph 1.3. I do not think the authors manage to properly convey the message. First of all, I would refer to volume rather than using expressions such as "Landslides that are small with respect to mountain size" or "For landslides that involve a large fraction of the mountain slope". The effect of volume on landslide mobility is indeed not fully understood and long runout landslides, which the authors clearly refer to, are characterized by volumes bigger than 1 million cubic meters. Important examples of**

terrestrial long runout landslides other than the Socompa rock avalanche exist, so I would extend the citations (e.g., Saidmarreh l., Blackhawk l., Heart Mountain l., Turtle Mountain l.). To present, many papers have try to contribute to the understanding of the influence of different parameters (volume, slope, gravity, fluid content, and so on) on landslide mobility. Just to name a few, McEwen (1989); Soukhovitskaya and Manga (2006); Lucas and Mangeney (2007); Lucas et al (2011); Johnson and Campbell (2017). Certainly this discussion is far from being close. For this reason I think that the sentence at Page 3 lines 29-32 ("This dependency of landslide propagation on slope, and indirectly on volume (and friction), initially identified on laboratory experiments (Farin et al., 2014), could be adequately documented by natural examples thanks to some very large Martian landslides (Borykov et al., submitted), much larger than any terrestrial landslide, which help populate the landslide dataset for voluminous landslides that propagate on nearly flat surfaces."), does not pay the right tribute to the papers that have already been published on long runout landslides and does not correctly report about the current knowledge on the influence of different parameters.

We would not like to extend too much this paragraph of Section 1.3 (now 1.4) because are landslides are only used as an example of a process which operates differently at different scales. However, we added the Blackhawk and Frank rock slides:

"For landslides that involve a large fraction of the mountain slope, the deposits spread in the nearly flat valley or plain downstream, such as in the case of the Socompa rock avalanche in Chile (e.g., Kelfoun and Druitt, 2005), the Blackhawk landslide in California, United States  (e.g., Shreve, 1987), and the Frank rock slide in Alberta, Canada (e.g., Daly et al., 1912)."

We also added McEwen (1989), Lucas et al. (2014) and and Johnson and Cambell (2017):

"It was shown that landslide volume is one of the factors that control landslide propagation (McEwen, 1989; Lucas et al., 2014; Johnson and Campbell, 2017) when the slope angle of the propagation plane is steeper than ca. 20º (Farin et al.; 2014; Borykov et al., submitted), whereas it has no influence for more gentle slopes. This dependency of landslide propagation on slope, and indirectly on volume (and friction), initially identified on laboratory experiments (Farin et al., 2014), could be adequately documented by natural examples thanks to some very large Martian landslides (Johnson and Campbell, 2017; Borykov et al., submitted)"

(note that Borykov is now being revised) instead of:

"Farin et al. (2014) and Borykov et al. (submitted) showed that landslide volume controls landslide propagation when the slope angle of the propagation plane is steeper than ca. 20º, whereas it has no influence for more gentle slopes. This dependency of landslide propagation on slope, and indirectly on volume (and friction), initially identified on laboratory experiments (Farin et al., 2014), could be adequately documented by natural examples thanks to some very large Martian landslides (Borykov et al., submitted)"

Soukhovitskaya and Manga (2006) focused on the role of water, Lucas and Mangeney (2007) on the role of the thickness of the destablised rock mass and Lucas et al (2011) on the geometry of landslide scars, which we think is too far for the purpose of this section.

**There is no mention on whether authors made the DEMs or not. If they did, I would mention what software they used (e.g., SocetSet or Ames Stereo Pipeline).**

The DEMs were indeed generated by one of the authors and with SocetSet®.

This information is added at the beginning of Section 2.1 (Data).

**Page 3 lines 2-5 "Uphill-facing normal faulting and crestal extensional deformation is indeed well documented on Earth in areas of DSGSD (review in Mège and Bourgeois, 2011). In most described terrestrial instances, such as in the Alps of Europe, Japan and New Zealand, the Andes and others, where DSGSD has been described in mountain ridges glaciated during the Quaternary (review in Mège and Bourgeois, 2011),.." – the use of "others" is not appropriate. Moreover, Mège and Bourgeois (2011) is not a review paper. More relevant papers about terrestrial cases can be cited.**

Mège and Bourgeois (2011) is not a review paper but includes a bibliographical review as supplementary material.

This is more explicitly indicated in the revised manuscript with "(see a review in the supplementary material of Mège and Bourgeois, 2011")

Many other references have been given in the new first section of he introduction; since this section is dedicated to Mars we prefer to keep the number of terrestrial references minimum. Others had been added because all the high mountains have DSGSD described and it would not make sense to list all of them here. In the revised manuscript "and others" has been removed.

**Page 3 lines 8-9 "..and additionally provides a good framework to understand the detected mineralogical occurrences as from CRISM (Mège and Bourgeois, 2011; Cull et al., 2014)." – Mège and Bourgeois (2011) do not work with CRISM data but they mention the identification of sulphates and hydrated silica covering the floor of Valles Marineris providing references. Even more recently, Watkins et al. (2015) provide evidence of hydrated silicate using CRISM data.**

Mège and Bourgeois (2011) and Cull et al. (2014) do provide a geological framework to understand the detected mineral occurrences, so we keep them and reorganise the text this way:

"… provides a good framework (Mège and Bourgeois, 2011; Cull et al., 2014) to understand the detected mineralogical occurrences as from CRISM (Roach et al., 2010; Cull et al., 2014)"

Watkins et al. (2015) appears more problematic for a number of flaws well pointed by Shaller (2016, doi: 10.1130/G37491C.1).

**Page 5 lines 6-7 "The observations of the Valles Marineris trough system reported here were done using Mars Reconnaissance Orbiter/CTX imagery as a baseline (Fig. 2)." – I would put a reference to Figure 1a and I would move it soon after Valles Marineris, removing "(Fig.2)".**

Fig. 1a has been added, and Fig. 2 removed (Figure 2 was removed anyway).

**Page 6 lines 2-3 "DEMs generated from CTX stereo pairs have the appropriate vertical precision of ca. 10 m,.." – Vertical precision of CTX stereo-derived DEMs is not so straightforward to assess. It is usually reported as several meter. Also, in Fig. 2, a different value is reported (? 15 m). Commonly in the literature, pixel resolution is given for CTX-DEM (~20 m).**

The precision of these DEMs is indeed difficult to assess; for the DEMs that we generated for this work it was estimated to 10-15 m.

Although Fig. 2 has been removed, we keep the conservative value of 15 m.

**Page 6 line 17 and Page 7 lines 1-2 "the faults cutting the profiles are located and marked as two lines illustrating two mean fault dip angles, $\alpha$=60° and $\alpha$=70° (Fig. 3), representative of unrotated shallow normal faults in extensional settings on Earth (e.g., Gudmundsson, 1992; Acocella et al., 2003). – I could not find any reference to fault dip angles in Acocella et al. (2003). In Gudmundsson (1992) I have found "Fault dip ranges from 42° to 89°, with a mean of 73°, but nearly 80% of the fault dip between 65° and 79°". I am not saying that the range 60° and 70° is not reasonable, I just do not see the justification in the literature proposed. Also, the authors clearly show that they are in favour of a DSGSD with a post-glacial origin scenario, almost discarding alternative explanations (Page 2 lines 20-22 "They denote extensional tectonics, but boundary forces that result in crustal "rifting" are unlikely to be the cause of this deformation even though rifting is frequently considered to have been a major contributor to the formation of some of the main Valles Marineris chasmata"). For this reason, I find slightly incoherent the reference to Gudmundsson (1992) and Acocella et al. (2003), who worked in the rift zone of Iceland and on flank instability of Mount Etna, respectively. Is there any fault dip range given for DSGSD in the literature?**

We would like to thank again Referee 2 for this thorough discussion on fault angles. We have not found any clear reference to fault angles in the DSGSD literature, most authors reporting on such scarps provide information about slopes, not dips. From our personal measurements, dips are indeed very difficult to

interpret because the fault planes themselves are in general covered by soil and vegetation. Minimum dips can be constrained only from the slope values, which qre very variable and concretely, of poor help. I checked again the Acocella et al. (2003) paper on the East African rift faults and must admit that the remembrance I had of it was proditory. Gudmundsson (1992) mentions a mean of 71 degrees fo the normal faults observed in the Tertiary lava pile, which is a more interesting value than those that can be measured at the surface, which in the basaltic setting of Iceland are very much influenced by columnar jointing geometry.

In the revised manuscript, we now justify the two values of 60 and 70 degrees this way:

> "The lower value corresponds to theoretical normal fault angles classically found in shear experiments and theory (e.g., Cloos, 1932; Anderson, 1951); the higher corresponds to normal fault angles that are commonly found in extensional tectonic regimes a few tens of metres below the surface (e.g., Gudmundsson, 1992), and could be similar to normal fault angles of DSGSD faults below the scarps. The 10º angle interval between 60° and 70° is considered as a plausible range of angular values, which cannot be retrieved from topography due to fault scarp erosion and debris accumulation."

instead of:

> ", …representative of unrotated shallow normal faults in extensional settings on Earth (e.g., Gudmundsson, 1992; Acocella et al., 2003)"

**Page 7 line 1 "..and marked as two lines illustrating two mean fault dip angles, $\alpha$=60$^\circ$ and $\alpha$=70$^\circ$ (Fig. 3).." – Figure 3 show just one line representing a general fault. So, I would remove the reference to Figure 3 at this line.**

Done.

**Page 7 lines 6-7 "The 10$^\circ$ angle interval is considered as the error on the true angular value, which cannot be retrieved from topography.." – I would place this sentence near to when fault dip range is discussed. This is the explanation to why the range 60$^\circ$-70$^\circ$ is considered.**

As explained just above, this is not the basic reason why these two values are selected, but we agree that this sentence had to be moved upward and so have we.

**Page 8 line 9 "Three sites [..] were selected in Valles Marineris, based on DEM gen- eration possibility,.." – Without DEM the study cannot be conducted, so not necessary.**

> We feel that mentioning this is necessary since as Referee 2 writes, this the basic criterion that makes our analyses possible. It needs to be said once.

**Page 10 lines 5-9 "Site 5 [..] is a ridge west of VeÄ¿ká Garajova Kopa on the way to VeÄ¿ká Kopa. [..] The 900 m wide ridge rises up to 250 m above the adjacent valleys. The height to width ratio R is 0.28." – These values correspond to Site 4 in Table 1 (??)**

The numbers in the text were wrong indeed. However, since this information is on Table 1 already we removed it from the revised text.

**Page 11 line 3 "topographic profiles were measured" –Profiles are traced. See also caption Fig.4.**

We have changed to "Measurements were done along 11 topographic profiles"

**Page 15 line 4 "Dh for the Valles Marineris sites (∼0.006) is roughly three times the values for the Tatras sites (∼0.002) only." – I did the calculation for Dh and I cannot obtain the same values. For terrestrial sites (4, 5, 6) I get ∼0.003, whereas for martian sites I get ∼0.008 if sites 1, 2b, 3 are taken; ∼0.008 if sites 1, 2a, 2b, 3 are taken; ∼0.007 if sites 1, 2a, 3 are taken; ∼0.006 if sites 1, 3 are taken. Please, explain how you did the calculation.**

We appreciate very much that Referee 2 took time to go into such details. We guess that he calculated the mean of Dh and Dv for each planet and taking the two boundary dip angles into account, which gives the result of 0.003 for the Tatras Mountains. For Valles Marineris, we considered the case where 1, 2a, 2b and 3

are taken into account, giving indeed 0.008. The ratio from Mars to Earth with these numbers is 8/3, and given the small number of ridges that were measured, we did not want to go into a value of more than 1 digit. This is why we wrote that the difference between Mars and Earth "is roughly three times".

In the revised manuscript, the values of Dh and Dv are appended to Table 2, which allows us to present a more rigourous analysis, giving in Section 4.2:

> "Dh for the Valles Marineris sites (0.005–0.010) is 2 to 2.6  times the values for the Tatras sites (0.002–0.005) only. Due to the steepness of normal faults, this difference is larger vertically, with Dv 2.6 to 5.1 larger for Valles Marineris (0.056–0.204) than for the Tatra Mountains (0.20–0.34)."

instead of:

> Dh for the Valles Marineris sites (~0.006) is roughly three times the values for the Tatras sites (~0.002) only. Similar difference occurs in Dv (~0.07 vs. ~0.02).

This also affects the discussion; after explaining why we favour the interpretation (2b) rather than (2a) for Melas Chasma, we can add this short penultimate paragraph in Section 5.2:

> "An implication of favouring interpretation (2b) is that the range of the difference between scaled horizontal and scaled vertical displacements for the studied Martian and terrestrial instances, already estimated in Section 4.2 from Table 2, is refined to 1.8–2.6 for Dh and 2.9–5.1 for Dv."

Similarly, the abstract was modified, with:

> "The measured horizontal spreading perpendicular to the ridges is proportionally 1.8 to 2.6 times larger for the Valles Marineris ridges than the Tatra Mountains, and vertically, 2.9 to 5.1 times, suggesting…"

instead of:

> "The measured finite strain of the Valles Marineris ridges is 3 times larger than in the Tatra Mountains, suggesting…"

**Page 16 lines 2-3 "This is interpreted as a consequence of removal of the highest part of the ridge, as discussed in Section 5." – I cannot find the discussion in Section 5.**

This is in Section 5.2., which was improved in response to an earlier comment (Specific Comment 3).

This issue is dealt with in the new second paragraph of this section:

> "Within the framework of a regional glacial landsystem, the northern side of the Melas-Candor ridge would naturally form by glacial erosion proceeding eastwards from a narrow graben located west of the ridge crest. The graben would widen and deepen, eventually resulting in a structurally controlled U-shaped valley and abrasion and the ridge crest."

The incriminated sentence in Section 4.2. was also improved:

> "This is interpreted as a consequence of glacial abrasion of the highest part of the ridge, as discussed in Section 5.2."

**Page 16 lines 11-12 "..the number of DSGSD faults in Valles Marineris (Tables 2 to 4) is not significantly different from the number of such faults in the Tatra Mountains." – No count of faults is given in any table. Also, there is not a Table 4.**

The number of faults is difficult to count because like for every fault system, fault scaling laws predict that their number is inversely proportional to their size, a feature which is clear in the field and is also apparent on some of the presented photographs (new Figure 5). We would therefore need to define (and justify) a fault size limit for counting for the terrestrial and Martian fault populations, considerably increasing the complexity of the discussion and moving us far from the scope of this work. This number issue is however not critical for this manuscript and we feel it safer to remove any reference to the number of faults.

We removed therefore "It is remarkable that in spite of much larger cumulated fault displacements (Table 3), the number of large DSGSD faults in Valles Marineris (Tables 2 to 4) is not significantly different from the number of such faults in the Tatra Mountains" at the beginning of Section 5.1, and "On both planets, strain is distributed over a small number (~5) of major scarps" in the abstract (which also solves the absence of Table 4).

**Page 16 line 13 "The very large fault offsets measured on individual faults in Valles Marineris require cumulated events (e.g., Fossen, 2010)." – Fossen (2010) is a textbook. If this has to be the reference, please provide the chapter and the page where the information can be found.**

This is almost common sense, resulting from crustal rheology and existence of rock strength; Fossen (2010) is one of the places (hence, "e.g.") where this is very simply explained. We added a reference to pages 172-174 of this book. Fossen (2010) is cited at the end of the same paragraph but the pages are the same and we do not feel that repeating these page numbers there is necessary.

COMMENTS ON FIGURES:

**Figure 2 – I think this is a superfluous figure.**

Agreed. Removed.

**Figure 3 – I would add a sketch/model showing which are the reference points from where the height and the width of a ridge are measured. For example, this information is always present in papers on long runout landslides, making clear that height drop and length of a landslide are measured from the highest point of the scarp to the furthest point of the deposit.**

We find this is an excellent idea. Figure 3 (now 2) has been complemented with such a model:

[Figure]

**Figures 4, 5, 6, 7 – I would create a border/space between the images.**

Done. Figure 5 was also split into two, following the Referee 1's suggestion. The quality of satellite imagery was also improved (as well as on Figure 7) and contours were added (new Figure 4). The field photographs were enlarged and make a new Figure 5.

**Figure 6 – I think in 6b and 6c codes of the area ("mc" and "co", respectively), then used in the name of the profiles in Fig.8a, are missing. Also, it took me a bit to find the codes "c1" and "c2" in 6a. I would create a box around the profiles.**

The size of "c1" and "c2" has been increased to make them easier to find on Figure 8, and the missing codes have been added to Figure 6. The font size of profile codes was also increased. The overall quality of the CTX image mosaic on Figure 6 was also much improved.

**Figure 8a, 8b – I think unit of measure should be placed on the graphs rather than in the caption.**

Moved.

COMMENTS ON TABLES:

**I would not use progressive numbers for site IDs, rather some other way that would help to identify immediately on the graphs which case is from which planet.**

We have renamed them M1, M2, M3, T1, T2 and T3.

**Table 2 – I suggest using the symbol x bar and z bar instead of writing "Mean hori- zontal fault displacement" and "Mean vertical displacement", respectively.**

Agreed. Changed.

**Table 2 – I suggest it to have the same style of Table 3. In Table 2, α is placed under Site ID, which does not make sense.**

Agreed. Changed.

**I would group Table 2 and 3 (and maybe also 1).**

This would lead to very long tables. Table 2 is now complemented with the values of Dh and Dv, which now already fills a line. The needs for Table 1 and Table 3 in the text are very far away, a reason for which we prefer to keep these tables separate.

**Values of Dh and Dv are not given in any table.**

They are now in Table 2.

COMMENTS ON SUPPLEMENTARY MATERIAL:

**I think that the data that the graphs are trying to show should be better made available in a table. On the graphs, the values of the displacement is not clear, making it difficult to follow the calculations that have brought to the results reported in the manuscript.**

The graphs were replaced by the corresponding tables in an excel file (Supplementary Material 1).

**I would add all the profiles obtained for this study, rather than just having few examples in the manuscript.**

Added as Supplementary Material 2, which is referenced in the results section 4.1:

"Examples of interpreted profiles are given on Fig. 8. Supplementary Material 2 includes all the profiles."

---

## Author Response (AR2)

**Response to Referee and Associate Editor**

Deep-seated gravitational slope deformation scaling on Mars and Earth: same fate for different initial conditions
By Olga Kromuszczyńska et al.

The response below mainly addresses issues raised by the Referee. The section "General comments", however, raises a point that has been also addressed by the Associate Editor, and can be viewed as an answer to the Associate Editor as well.

**Note**

This is a review of the second, revised version of the manuscript. The original version has been reviewed by two anonymous referees (https://dio.org/10.5194/esurf-2018-27-RC1 and https://dio.org/10.5194/esurf-2018-27-RC2), and I have seen their comments and a detailed, point-to-point response letter by the authors. I have not seen the original version of the manuscript as it was first submitted.

**Overview**

The authors describe morphometrically two cases of Deep-Seated Gravitational Slope Deformation (DSGSD), one in the Tatra mountains (Earth), and one in the Valles Marineris (Mars). The former is known to be a result of post-glacial destabilization by slope-debuttressing after removal of glacial ice, a well-known mechanism which is also thought to be responsible for the latter. Morphologically, DSGSD is characterized by normal faulting which produces uphill-facing scarps, ridge-top splitting, and summital valleys. The authors performed measurements of scarp offsets along topographic profiles oriented normal to the fault strike, i.e. parallel to the extension. The total height (H) and width or length (L) of the DSGSD-affected ridges was also measured, and their ratio R (i.e. H/L) determined. The authors find that the R value is similar for the two investigated cases of DSGSD and propose that the R value may be considered a measure for "maturity" of the process, i.e. that DSGSD-affected ridges will assume a given R value (~0.24) when DSGSD has ended. This proposal is based on the assumption that DSGSD is not active anymore at the investigated ridges on Earth and Mars, and that the horizontal and vertical deformation was proportionally different, i.e. the original ridge geometry was different.

**General comments**

The manuscript has already been revised according to critical comments by two reviewers, and it is in good technical shape. The language is clear, the illustrations are well prepared and legible, figure captions contain the necessary information, and referencing is up-to-date. Overall, the topic should be of interest to readers of Earth Surface Dynamics. As such, the manuscript might be considered for publication. However, I share one of the main concerns of the second reviewer of the original version (https://dio.org/10.5194/esurf-2018-27-RC2), and I am not fully convinced be the author's response. This concern is about the essential take-home message of the manuscript (if it is stripped to its bones), i.e. the proposed use of the R value as a measure of slope maturity of ridges that are subject to DSGSD. Only two study sites are investigated, and I would concur with the previous reviewer #2 that (many) more such sites, especially on Earth but also on Mars, should be investigated before such a conclusion can be made. The response by the authors to this concern is basically only some rewording, but they did not analyze more study sites. In the opinion of this reviewer, it is completely

unclear if the similar R-values in the Tatra mountains and in Valles Marineris are only a coincidence, or if they are indeed an indication of the final state of DSGSD-affected topographic ridges. Only more data could help, and I would strongly recommend that the authors collect such data to support (or reject) their weakly based hypothesis.

We agree that more data would strengthen our results, and this is apparent from the conclusion that with this dataset we do not intend to draw general conclusion:

> Individual fault displacements across DSGSD scarps in the Tatra Mountains are similar to fault displacement in most DSGSD 15 sites on Earth (see references in the Supplementary Table 1 of Mège and Bourgeois, 2011), suggesting that this conclusion may be extrapolated to other regions. Nevertheless, similar analyses need to be conducted in other ridges affected by DSGSD, formed in post-glacial as well as non-postglacial conditions, both inactive and active, before general conclusions can be drawn.

We thank the Associate Editor to acknowledge this point:

> "We do note your response that it is a provisional finding that needs to be tested"

but will argue that the following statement:

> "However, we contend that the 'testing' is within the scope of this manuscript and is required in order for it to be accepted for publication."

may not be shared by us for several reasons.

First, the terrestrial examples are actually two In the current manuscript, the lower and upper Tatras being distinct structural and geomorphologic ensembles.

A major constraint for testing is the availability of adapted topography data, i.e., having a vertical precision better than 1 m. This precision is higher than the precision offered by conventional topographic maps, and far higher than more recent datasets obtained in satellite missions such as SRTM and ASTER. In the absence of DSGSD profiling surveys through the entire slope of ridges by other authors, as far as we know, the only way to obtain relevant additional data would consist in building a new project targeting another terrestrial analogue in another mountain belt (including funds for field expenses, and manpower for several months to analyse data (three of the co-authors of this manuscript have left academia, including the first author since manuscript submission).

We realise that Referee 2 may not have fully appreciated the amount of work having made this manuscript possible, estimating that "the scientific content is limited to a few topographic measurements". We guess that this statement has heavily influenced the request for adding additional data from other mountain areas. Each ridge profile corresponds to thousands of measurements with a vertical precision of 40 cm; perhaps the feeling that the number of measurements is small is coming from that this precision is not given as elevation points with 20 cm of errors on both vertical sides, the instrument is configured in such a wat that a new value is recorded each time the sensor has vertically moved by 40 cm. For instance, the number of elevation measurements at Site 6 is 7064 (see the measurement density on the first figure below).

Not only is the number of measurements high, but the technology used for these measurements is novel in earth sciences, as explained in Kromuszczyńska et al. (2016). The Wide Area Differential GPS technology, which is for instance routinely used during aircraft landing, had never been used in earth sciences to our knowledge, which made necessary a full protocol of evaluation, data analysis, and validation using data obtained during the same two field seasons as the data used in the work behind this manuscript. As indicated in Kromuszczyńska et al. (2016), the reason why the WADGPS technology has been used is that it allows to obtain a vertical precision (depending on the atmospheric conditions) which is enough for many geomorphological analyses, with little volume and weight to carry, and

without the need to proceed to very long-lasted measurements at fixed stations. Data can be in theory obtained during a walk, a complicating factor being the state of the atmosphere. Atmospheric corrections are performed in real time, but atmospheric instability is less well corrected than when the atmospheric pressure is stable between the GPS satellites and the receiver. Due to the number of data as well as the variable atmospheric corrections, then to additional corrections such as for transforming irregular paths (due to e.g. shrubs on the way) to profiles perpendicular to fault scarps, each profile represents a fair amount of work. The work behind this paper is far from having a walk and withdrawing data to an excel file.

Due to the amount of work behind this manuscript, the authors feel that requesting insertion of new work and testing would be as if, for instance, the authors of a study of deltaic processes on Mars with a field-based comparison with the Mississippi delta, would be requested at the revision stage to perform additional field work in the Nile or Ganges delta to check that the conclusions reached using the Mississippi also applies to other deltas. In the present manuscript, the Tatra Mountains study already is the outcome of two field work seasons and several tens of thousands of measurements. In summary, the data we are presenting here cover a huge piece of work and are scientifically strong; the quantity of data and work behind does justify, in our opinion, a full paper. Confirmation by additional field campaigning will be the most welcome to broaden the conclusions made here, but is not mandatory to support the article conclusions.

On Mars, only CTX and HiRISE DTMs can provide the appropriate vertical resolution. However, there is no HiRISE stereo coverage adapted to the study of DSGSD transects across ridges, and only a small number of CTX images (the acquisition of which has not been specifically designed for stereophotogrammetry) can be used for stereo in these areas, most of which (and all those where DSGSD is well geomorphologically characterised) are studied in the manuscript. There is little potential of improvement with the orbital data of today; this has just been checked using the newest CTX image database available today. HiRISE stereo images can be requested for acquisition through HiWish; but Valles Marineris is the busiest place on Mars for HiRISE image acquisition. Stereoscopic coverage additionally requests specific image acquisition parameters. Each specified geometric parameter decreases the likelihood of having the appropriate image obtained in a reasonable time; similarly, the obtained HiRISE stereo images would need to cover a full ridge section, meaning a very large number of downlinked bits and an increased acquisition time. By experience, should any suitable stereo pair be acquired by HiRISE, there is little chance that it would happen before 2 years, probably longer, if ever.

Therefore, we do believe that after taking all the precautions to obtain the best possible measurements, both on Earth and Mars, the data that we present are of a quality that justifies the conclusions of the manuscript. Study of any further region is undoubtedly desirable, especially aiming at broadening our conclusions, but we do not think that this paper is the appropriate place.

**Other comments**

Section 1.1: This is certainly interesting information, but most of it (e.g., the compressional bulging, the decollements, etc) is not really linked to any later part of the manuscript. I still find (following the comment#1 of reviewer #1) that the introduction does not clearly outline the problem and explains the author's approach. Section 1.1 (newly added) is somehow "glued" to the front of the manuscript, but does not form a consistent explanation of why all this is relevant for the reader.

With hindsight, we fully agree with this comment and tried to improve this section in the new manuscript. The new section 1.1 has been rephrased to provide a better explanation why this work has been undertaken, and a long sentence that was suddenly going into details of basal bulging and overthrusting, which were of limited use for the remaining of the manuscript, was removed. We have kept mention of bulging and overthrusting because it is sometimes a significant component of DSGSD,

and show how we implicitly deal with this type of deformation (this information was given in the conclusion in the former manuscript). Thanks to this new section, the introduction is now better connected to the scope of the manuscript.

Page 5, lines 7-8: "DSGSD scarps were studied in the Tatra mountains during two periods of field work." This sentence is out of context here – I suggest moving it to the Data section (2.1).

Moved to the beginning of the Data section.

Page 5, lines 21-24: It is very speculative (and not "likely") if "liquid water in a melting mountain permafrost" ever played a role on Mars, let alone in a specific environment such as Valles Marineris. On what observation (or published study) is this notion based? Moreover, permafrost thaws, it does not "melt" (this mistake is made very often, mostly when permafrost is confused with ground ice: permafrost is defined by temperature only, so it thaws - ice melts).

We acknowledge that demonstration was made that the environmental conditions changed, whereas demonstration of changes in the resulting processes has not. The original text:

> Development of DSGSD features in Valles Marineris is therefore likely to have proceeded in response to a variety of processes recurrently operating over a very long history; both short-term processes such as ridge postglacial debuttressing, and long-term evolution under the influence of liquid water in a melting mountain permafrost (e.g. Noetzli and Gruber, 2009; Huggel et al., 2013).

was changed to:

> Development of DSGSD features in Valles Marineris is therefore likely to have proceeded under a long succession of contrasted environmental conditions, perhaps resulting in a variety of processes operating recurrently, both short-term processes such as ridge postglacial debuttressing, and long-term evolution under the influence of liquid water in a thawing mountain permafrost (e.g. Noetzli and Gruber, 2009; Huggel et al., 2013).

Page 9, line 3-4: I don't understand this. Please explain more clearly.

This sentence was indeed confusing in addition to being redundant with the previous explanations. It has been removed.

Page 11, line 13: Normal fault dips are assumed to be 60° to 70°, and normal faults are mode II fractures, yet here the authors talk about tension fractures (mode I fractures). This seems like a contradiction – I suggest the authors clarify the respective role of tension fracturing and shear fracturing in DSGSD.

Page 16, line 10: If DGPS is used (page 7, lines 12-13), a higher vertical precision than 40 cm should be achievable. It is also unclear how dense the horizontal sampling distance was during field work. The profiles in Figure 8b do not really look that great – from visual inspection of the left part of the profile at site T2, it seems that measurements were only made every few meters (note the long perfectly horizontal section of the profile between distances of ~4 m and ~9 m).

I miss a discussion of the potential implications of DSGSD on the thermal nature of Martian glaciers. If DSGSD requires slopes to be oversteepened (so they get unstable after debuttressing by melting or sublimating of ice), this implies glacial erosion of slopes. Typically, glacial erosion is effective for polythermal or "warm-based" glaciers, while cold-based glaciers are frozen to their beds and do not

cause significant erosion (but some erosion is also caused by cold-based glaciers). If this consideration is correct, the observation of DSGSD on Mars would potentially imply warm-based glaciers, a contradiction to most models which predict that glaciers on Mars in most climatic scenarios should be cold-based. The authors are encouraged to elaborate on this possibility - although it is probably beyond the original scope of this manuscript, it may add some "beef" to it. As it is now, the scientific content is limited to a few topographic measurements and a weakly based speculation (the use of R). More scientific discussion would add to the value of the manuscript.

We refute the statement that "the scientific content is limited to a few topographic measurements and a weakly based speculation (the use of R)", given that tens of thousands of measurements points have been accounted for in total, as discussed at the beginning of this response. The figure below is a screenshot of GPS profiles measured along downslope profiles at Site 6, given as an example. The white mountain trail at the top of the topographic ridge provides a visual scale; the GPS data points are dense enough to define an almost continuous line. Vertical precision, resulting from the WADGPS technique described in Kromuszczyńska et al. (2016), is 40 cm, which is why some portions of profiles appear perfectly flat over several metres. If terrain topography is not varying by more than 40 cm, the recording GPS elevation will stay constant.

[Figure]

DSGSD does not require slopes to be oversteepened; in the revised Introduction (Section 1.1, second paragraph) we stress that ridge core crushing may replace lower ridge deformation, with the appropriate references (observation reports and models). In Valles Marineris, whether bulging has occurred is difficult to assess due to the widespread debris slopes (as indicated in the same paragraph). Therefore, a discussion on whether warm-based glaciers or cold-based glaciers were present would not be based on strong grounds. Furthermore, wall oversteepening and erosion might also be expected in case of cold-based glacier to do ice sublimation, but sublimation rate and how it would proceed geometrically is not better constrained. Such an interesting discussion would quickly turn to a nightmare that would lead to weak conclusions, in addition to – as noted by the Referee – bringing us far from the manuscript scope.

Page 21, line 11-12: Is it not possible to determine rather than inferring the initial slopes of Valles Marineris walls? Maybe the authors can make some measurements elsewhere in VM, where DSGSD has not affected the slopes.

Such areas are not easily found, and actually do not help because slopes without DSGSD are not "initial". We explain below that since DSGSD in Valles Marineris is thought to be Upper Hesperian to Lower Amazonian, "initial" slopes are expected to have failed; the slopes who have not failed were simply too gentle to be unstable.

Areas not affected by DSGSD are not observed around sites M2 and M3 (as well as in most DSGSD sites in Valles Marineris not covered by this study due to unavailability of DTM data with sufficient vertical precision), where the whole ridge is similarly deformed by DSGSD. At the ridge where Site M1 is located, however, the whole ridge has been deformed as well, but much less east of Site M1. In that eastern area, only one DSGSD scarp developed (red profile on the figure), and let us use it as a proxy for calculation of potentially "initial" slope angles.

The table below gives the averaged slope angles along two deformed ridge profiles in the western part of Site M1, and along a profile measured on the almost undeformed ridge. At Site M1, DSGSD is observed on the northern slope, which has the highest values, 21-23°. The slope angles of the undeformed ridge are 16 to 19°, which simply shows that these slopes were not steep enough to fail. The initial slopes of the failed areas can therefore not be inferred from areas nearby because those areas do not display "initial" slopes, instead they show more gentle slopes which did not fail because they are stable.

| Deformed ridge (Site M1, west) | | Almost undeformed ridge | |
|---|---|---|---|
| Northern slope (with DSGSD) | 21-23° | Northern slope (with DSGSD) | 16° |
| Southern slope (without DSGSD) | 15-18° | Southern slope (without DSGSD) | 19° |

[Figure]

Themis daytime image mosaic with HRSC DTM contours, vertical spacing 500 m. See explanation above.

Page 1, lines 27-29: The last sentence of the abstract is largely redundant (cf. lines 21-22) and could be deleted.

Deleted

Figure 2: I would appreciate if the compressive bulging (page 2, line 6) is illustrated in Figure 2a.

Added

Figures 8a and 8b: If I understood the text correctly, the term "reconstruction" is not really appropriate. The fault traces are just plotted where the slope suggests their position, and the dips are actually assumed, not "reconstructed".

Agreed; replaced by "Determination"

Figure 9: What is the vertical extent of the grey bars? I understand that it represents the range of values as given in Table 2 (?). But if this true, why is the range of Dh for site 2(a) from 0.007 to ~0.12 (estimated from Figure 9a), whereas Table 2 gives values for Dh for site 2(a) of 0.010 (a=60°) and 0.007 (a=70°)? Is there a typo in the table (0.010 instead of 0.11)? I also can't see that the values for the Tatra sites are between 0.002 to 0.005 – from Figure 9a, I would estimate values between ~0.02 and ~0.03. Same for Dv: Why is a values for VM (0.056-0.204) larger than the values for the Tatra mountains (0.20 – 0.34)? To me, it looks smaller. I may have missed the point in comparing Table 2 with Figure 9 – any clarification would be helpful. This was already a source of confusion for previous reviewer #2, and I strongly recommend to double-check the numbers.

We are grateful to the Referee to point this inconsistency, and would like to apologise for the time spent to try to understand the erroneous Figure 9. After checking the original files and calculations, Table 2 is correct, the corresponding discussion is correct, but figures 9a and 9b were wrong. Figure 9a had wrong numbers reported for Site 2a, and the scale of Figure 9b was erroneously labelled from 0 to 0.02 instead of 0 to 0.2. The new Figure is now fully consistent with Table 2 and the text.

Page 22, line 5: Laskar et al. (2004) is not an appropriate reference here.

This shortcut was removed:

> Fault reactivation may have occurred as a geologic response to the long succession of glacial/interglacial cycles expected to have occurred throughout the history of Valles Marineris (Laskar et al., 2004; Mège and Bourgeois, 2011; Gourronc et al., 2014).

has been changed to:

> Fault reactivation may have occurred as a geologic response to the long succession of glacial/interglacial cycles expected to have occurred throughout the history of Mars from celestial mechanics (Laskar et al., 2004) and in particular, in Valles Marineris (Mège and Bourgeois, 2011; Gourronc et al., 2014).

**Formal issues**

We thank the Referee for this very careful and helpful manuscript examination.

Page 1, line 19: replace "two orders" by "two orders of magnitude"

Done

Page 1, line 26: "Martian obliquity"

Done

Page 1, lines 25-26: "activity of faults" – does a landform (a fault) have an activity? I would suggest writing "activity of the faulting", because a process (faulting) does have duration of activity.

We do not share the view that a fault is a landform (in contrary to a fault scarp), and rather define it as a structural object. "Fault activity" is widely used in active tectonics, i.e. when faulting is connected to seismic activity, for many years (e.g. to describe activity of the famous Wasatch fault in Utah, Cluff et al., 1975, Tectonophysics 29, 161-168), and is used in the best textbooks, such as Structural Geology (H. Fossen, 2010) and The Mechanics of Earthquakes and Faulting (C.H. Scholz, 2002). A short Google search today confirms that "activity of the faulting" gives 9 results, 'faulting activity" 7,980, and "fault activity" 126,000.

Page 3, line 24: remove blank after DSGSD.

Done

Page 3, lines 24-30: Something is wrong with the sentences here (a period seems to be missing, splitting this long text into two or more sentences). Please check and rephrase.

The problem was due to the presence of "where", (previously in line 24) which is now removed.

Page 3, line 31: remove comma after "retreat"

Done

Page 4, line 10: replace "fully similar" by "self-similar" (?).

> For instance, landslide propagation does not depend on fully similar parameters when small and large.

has been replaced by:

> For instance, landslide propagation does not critically depend on the same parameters when small or large.

Page 5, lines 15-16: "some instances […] sometimes" remove "sometimes"

Done

Page 5, line 17: "are not against" this reads awkward. Perhaps reword like "[…] based on observations by Mège and Bourgeois (2011) it is not possible to exclude"

Done

Page 7, line 3: replace "done" by "made"

Done

Page 11, line 15: replace "with" by "by"

Done

Page 18, line 9: Add comma after "displacement Dh (a)"

Done

Figure 6: Th caption says the fault scarps are marked by black lines, however, in the figure they are red.

Done

Page 16, line 2: "[…] graben at, e.g., Site M1"

Done

Table 2: The lines for Dh and Dv are offset with respect to lines for x(m) and z(m).

Aligned

Figure 9: On the x-axis, write site "M1"; "T4" etc. instead of just "1", "4", etc.

Done

Page 18, line 10: "intervening" reads awkward. Please try to reword.

Replaced by "central"

Page 19, line 4: "gravitational acceleration"

Done

Page 19, line 14: replace "much deep" with just "deep"

Done

Page 21, Table 3, 1st line: "ex" is offset with respect to the other columns.

Aligned

Page 21, line 13: "does not carry an indication" (insert "an")

Done

Page 21, line 14: "scarps" (plural)

Done